# A General Spatio-Temporal Backbone with Scalable Contextual Pattern Bank for Urban Continual Forecasting

**Aoyu Liu, Yaying Zhang**[*]

The Key Laboratory of Embedded System and Service Computing, Ministry of Education,
Tongji University, Shanghai, China
`{liuaoyu, yaying.zhang}@tongji.edu.cn`

## Abstract

With the rapid growth of spatio-temporal data fueled by IoT deployments and urban infrastructure expansion, accurate and efficient continual forecasting has become a critical challenge. Most existing Spatio-Temporal Graph Neural Networks rely on static graph structures and offline training, rendering them inadequate for real-world streaming scenarios characterized by graph expansion and distribution shifts. Although Continual Spatio-Temporal Forecasting methods have been proposed to tackle these issues, they often adopt backbones with limited modeling capacity and lack effective mechanisms to balance stability and adaptability. To overcome these limitations, we propose `STBP`, a novel framework that integrates a general spatio-temporal backbone with a scalable contextual pattern bank. The backbone extracts stable representations in the frequency domain and captures dynamic spatial correlations through lightweight linear graph attention. To support continual adaptation and mitigate catastrophic forgetting, the contextual pattern bank is updated incrementally via parameter expansion, enabling the capture of evolving node-level heterogeneity and relevance. During incremental training, the backbone remains fixed to preserve general knowledge, while the pattern bank adapts to new scenarios and distributions. Extensive experiments demonstrate that `STBP` outperforms state-of-the-art baselines in both forecasting accuracy and scalability, validating its effectiveness for continual spatio-temporal forecasting. Code is available at https://github.com/Aoyu-Liu/STBP.

## 1 Introduction

With the rapid development of urban IoT sensing systems, spatio-temporal data such as traffic flow (Shao et al., 2022b) and air quality (Tian et al., 2025) observations continue to surge (Kumar et al., 2024; Hu et al., 2023; Fang et al., 2026). Conducting efficient and accurate forecasting on data streams has become a core task in the development of smart cities (Jin et al., 2024; Yang et al., 2025). Unlike traditional offline learning based on static assumptions, real-world urban environments are in a state of continuous evolution—dynamic changes in urban structure and behavioral patterns constantly drive the evolution of graph structures and data distributions.

Spatio-Temporal Graph Neural Networks (STGNNs) (Kong et al., 2024; Gao et al., 2024; Liu & Zhang, 2025) have been widely used to model complex spatio-temporal dependencies. However, most existing models still adhere to the paradigm of "fixed topology + offline training": the graph structure is predefined and fixed during the training phase, and the model is deployed directly after training. Yet, as shown in Figure 1, this static assumption becomes difficult to sustain when the node set continuously expands or the connectivity dynamically reconstructs over time. If one relies solely on structural modifications and continuous fine-tuning to handle node increments, model performance often degrades significantly. Therefore, Continual Spatio-Temporal Forecasting (CSTF) (Miao et al., 2024b; Chen & Liang, 2025; Ma et al., 2025b) has garnered increasing attention. Its goal is to achieve

---

[*]Corresponding author.

incremental learning and efficient inference on new data without repeatedly relying on retraining with historical data. As shown in Figure 1, typical CSTF approaches employ a general spatio-temporal backbone integrated with strategies such as regularization, replay, or dynamic architectures to adapt to graph structural expansion and mitigate catastrophic forgetting.

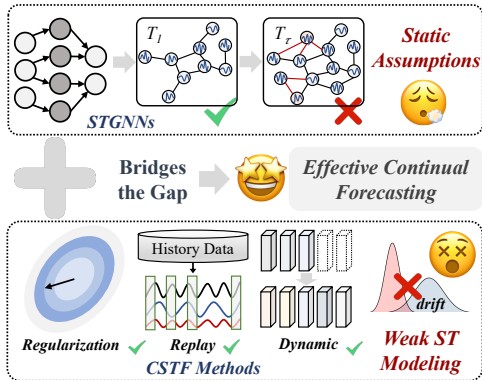

Figure 1: Limitations of existing studies.

However, two key issues in existing CSTF methods have not yet been adequately addressed. First, the general backbone adopted by most current methods is relatively simple (e.g., stacks of graph and temporal convolutions), making it difficult to effectively handle incremental scenarios characterized by dynamically changing spatio-temporal correlations and long-term distribution drift. Forcibly adapting existing STGNNs for continual learning often leads to performance degradation (Shao et al., 2024; Ma et al., 2025a). Second, continual optimization strategies based on dynamic structural expansion are often weakly coupled with the backbone—such as direct parameter expansion or prompt concatenation—making it challenging to achieve a good balance among model stability, adaptability, and interpretability. Based on the above issues, we argue that an ideal CSTF framework should simultaneously address the following four key challenges: ❶ *handling distributional drift;* ❷ *modeling dynamic spatio-temporal correlations;* ❸ *alleviating catastrophic forgetting;* and ❹ *designing an incremental strategy that efficiently collaborates with the backbone.*

To this end, we bridge the gap between STGNNs and continual learning by introducing a general-purpose **s**patio-**t**emporal **b**ackbone with scalable contextual **p**attern bank (`STBP`). Specifically, the backbone in `STBP` leverages frequency-domain modules to extract stable spatio-temporal components, mitigating distributional drift. Simultaneously, a lightweight, scene-agnostic linear graph attention mechanism is introduced to model dynamic spatial correlations with low computational overhead. To mitigate catastrophic forgetting and support continuous graph structure expansion, we design a contextual pattern bank composed of trainable parameters. It incrementally updates knowledge via parameter expansion and interacts with the backbone through gating and attention mechanisms, thereby uncovering node relevance and heterogeneity, and gradually adapting to scenario expansion at low cost. Within this framework, the backbone is responsible for modeling general and stable patterns, while the contextual pattern bank captures node-related heterogeneous contexts, working collaboratively to adapt to continuously evolving environments.

Our main contributions are summarized as follows: ❶ We propose an efficient and general backbone tailored for continual forecasting tasks, capable of modeling dynamic spatial correlations and mitigating distribution shift; ❷ We design a prompt-based guidance mechanism using contextual pattern bank, supporting dynamic model adaptation and alleviating catastrophic forgetting; ❸ Extensive experiments on multiple real-world datasets demonstrate that `STBP` significantly outperforms state-of-the-art baselines in terms of forecasting accuracy, adaptability, and scalability.

## 2 RELATED WORK

**Spatio-Temporal Forecasting.** Early studies in spatio-temporal forecasting, including methods like STGCN (Yu et al., 2018) and DCRNN (Li et al., 2018), primarily focused on combining basic temporal and spatial elements for prediction tasks. These models typically depended on predefined geographic adjacency matrices, which limited their ability to capture the evolving nature of spatial correlations. In contrast, later advancements, such as GWNet (Wu et al., 2019), DGCRN (Li et al., 2023), and MegaCRN (Jiang et al., 2023b), addressed this limitation by incorporating adaptive adjacency matrices or learning spatial correlations directly from the data. This shift led to a notable improvement in forecasting accuracy. More recently, models like STID (Shao et al., 2022a), STAEformer (Liu et al., 2023a), and HimNet (Dong et al., 2024) have emphasized the significance of distinguishing spatial patterns to further enhance forecasting performance. These methods incorporate trainable components, including spatial embeddings, parameter pools, and contextual pattern bank, to more accurately capture spatial variations, boosting both prediction precision and model adaptability.

**Continual Spatio-Temporal Forecasting.** TrafficStream (Chen et al., 2021), one of the pioneering frameworks in CSTF, integrates spatio-temporal modeling with continual learning by employing historical data replay and parameter smoothing to manage long-term streaming traffic data and achieve accurate traffic flow prediction. Building on this line of work, STKEC (Wang et al., 2023a) proposes an influence-based knowledge expansion strategy together with a memory-augmented knowledge consolidation mechanism, which better supports the scaling of transportation networks while alleviating catastrophic forgetting. PECPM (Wang et al., 2023b) leverages pattern matching to dynamically maintain a traffic pattern bank, enabling efficient, historical-data-free continual learning with improved accuracy. STRAP (Zhang et al., 2025) adopts retrieval-augmented learning, constructing multi-dimensional pattern libraries and using plug-and-play prompting to fuse retrieved patterns, thereby enhancing out-of-distribution (OOD) generalization and mitigating catastrophic forgetting. EAC (Chen & Liang, 2025) introduces prompt tuning via a dynamic prompt pool that expands and compresses over time, balancing adaptation to new nodes with knowledge preservation in a parameter-efficient manner. Additionally, UFCL (Miao et al., 2025) leverages federated learning to protect data privacy and employs a global replay buffer of synthetic spatio-temporal data, addressing the challenges of distributed streaming environments.

## 3 Preliminary

**Definition 1 (Streaming Spatio-Temporal Graph).** We define a streaming spatio-temporal graph as a sequence of evolving graphs $\mathbb{G} = \{G_\tau\}_{\tau=1}^{\mathcal{T}}$, where each graph $G_\tau = (V_\tau, E_\tau, A_\tau)$ represents the graph at incremental period $\tau$. Here, $V_\tau$ denotes the node set, $E_\tau$ the edge set, and adjacency matrix $A_\tau \in \mathbb{R}^{N_\tau \times N_\tau}$ connections between nodes. The number of nodes at period $\tau$ is denoted by $N_\tau = |V_\tau|$. The graph evolves incrementally as $G_\tau = G_{\tau-1} + \Delta G_\tau$, where $\Delta G_\tau$ captures structural or feature modifications between periods.

**Definition 2 (Continual Spatio-Temporal Forecasting).** Continual spatio-temporal forecasting aims to develop an optimal predictive model at each stage based on dynamic, streaming spatio-temporal graph data. At each incremental period $\tau$, given the current graph $G_\tau$ and historical observations $\mathbf{X}_\tau \in \mathbb{R}^{N_\tau \times T_h}$, the goal is to predict future signals $\mathbf{Y}_\tau \in \mathbb{R}^{N_\tau \times T_f}$ as follows:

$$\hat{\mathbf{Y}}_\tau = f_\theta(G_\tau, \mathbf{X}_\tau), \tag{1}$$

where $T_h$ is the length of the historical observation window, and $T_f$ is the forecasting horizon. The model $f_\theta$ is parameterized by $\theta$, and continually updated by minimizing:

$$\theta_\tau^* = \arg\min_\theta \mathbb{E}_{(G_\tau, \mathbf{X}_\tau, \mathbf{Y}_\tau) \sim \mathcal{D}_\tau} \left[ \mathcal{L}\left(f_\theta(G_\tau, \mathbf{X}_\tau), \mathbf{Y}_\tau\right) \right], \tag{2}$$

where $\mathcal{L}(\cdot, \cdot)$ is a loss function, and $\mathcal{D}_\tau$ denotes the data distribution at period $\tau$.

## 4 Methodology

### 4.1 Overview of STBP

The workflow and architecture of STBP are shown in Figure 2. It consists of two core components: a general spatio-temporal backbone and a contextual pattern bank. The backbone, comprising temporal and spatial modules with a prediction layer, captures spatio-temporal correlations in evolving networks. The contextual pattern bank, made of trainable parameters, is dynamically expanded and fine-tuned as data evolves. While the backbone captures general, stable patterns, the contextual pattern bank adapts to environmental changes, focusing on context-specific patterns. Guided by prompts, both components collaborate to form an efficient and robust continual learning system.

In terms of workflow, streaming spatio-temporal data is sequentially fed into the STBP. During the initial incremental training phase, the backbone and contextual pattern bank are jointly trained to capture spatio-temporal correlations from current data. In later stages, the backbone is frozen (denoted by a snowflake) to retain knowledge learned from historical data, while the contextual pattern bank is updated (denoted by a flame) through expansion and fine-tuning. These updates serve as prompts, guiding the frozen backbone to adapt to new data distributions. This continual learning process, driven by the interplay between backbone and contextual pattern bank, enables the model to progressively enhance its representation power and adaptability while preserving core functionality. For detailed workflow steps, refer to Algorithm 1 in Appendix A.3.2.

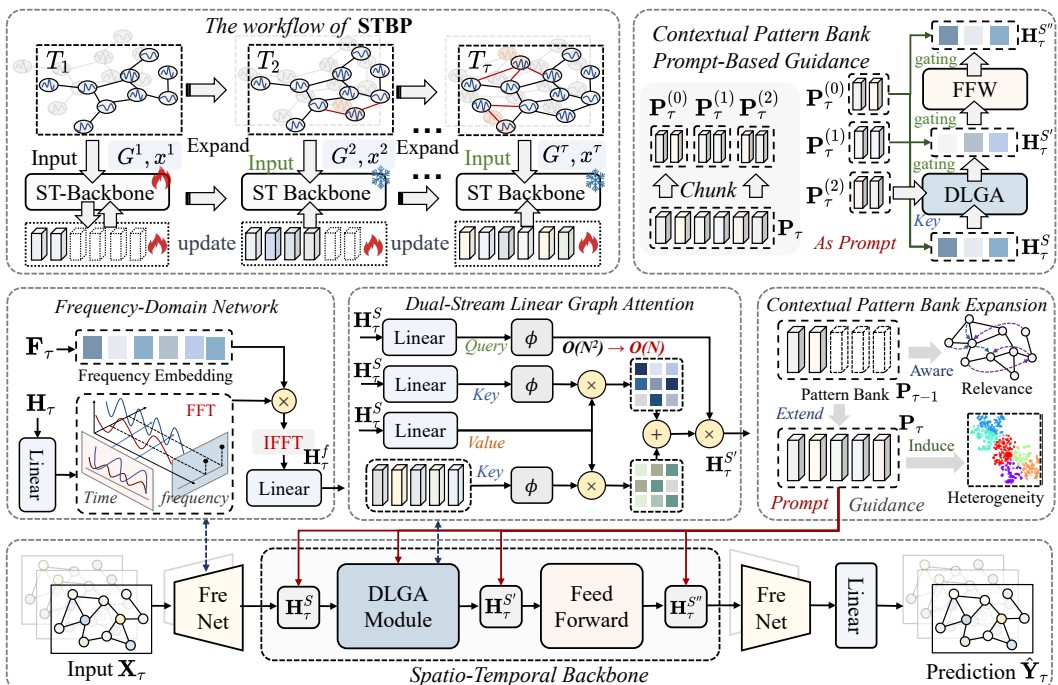

Figure 2: The overall workflow and architecture of STBP.

## 4.2 CONTEXTUAL PATTERN BANK

Recent studies (Shao et al., 2022a; Dong et al., 2024; Chen & Liang, 2025) have shown that incorporating node-specific trainable parameters into STGNNs can significantly enhance forecasting performance. Following this insight, we propose an expandable `contextual pattern bank` $\mathbf{P}_\tau \in \mathbb{R}^{N_\tau \times d}$, composed of trainable parameters, to consolidate historical spatio-temporal patterns and generalize to new ones, thereby mitigating *catastrophic forgetting* and continuously adapting to new incremental scenarios, where $d$ denotes the feature dimension.

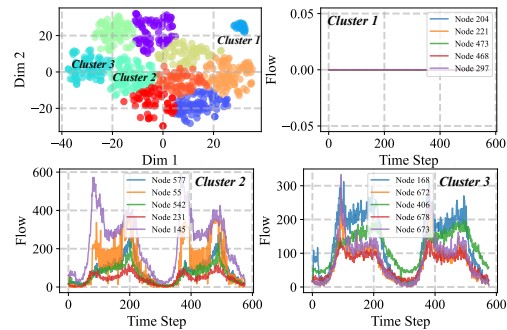

Figure 3: Contextual pattern bank visualization.

We posit that the model can utilize $\mathbf{P}_\tau$ to effectively distinguish both the *relevance* and *heterogeneity* of nodes, enabling a more nuanced understanding of the underlying data structures. Here, *relevance* refers to shared behavioral patterns among nodes—such as similar trends or periodic fluctuations—while *heterogeneity* captures differences arising from distinct node functions or external factors such as geography, policy, or events. To validate this hypothesis, we conduct a t-SNE-based analysis on $\mathbf{P}_\tau$ trained on spatio-temporal datasets (see Figure 3), which reveals meaningful clustering patterns. Each cluster exhibits distinct characteristics, corresponding to *heterogeneity*, while nodes within the same cluster display similar temporal dynamics, reflecting *relevance*.

As shown in Figure 2, given a streaming spatio-temporal input $\mathbf{X}_\tau \in \mathbb{R}^{N_\tau \times T_h}$, the backbone model $\mathcal{M}_\theta$, and contextual pattern bank $\mathbf{P}_\tau \in \mathbb{R}^{N_\tau \times d}$, the incremental learning process is formulated as:

$$\hat{\mathbf{Y}}_\tau = \mathcal{M}_\theta(\mathbf{X}_\tau, \mathbf{P}_\tau). \tag{3}$$

At the initial training stage ($\tau = 1$), both the backbone and contextual pattern bank are jointly trained (denoted with flame). For subsequent stages ($\tau > 1$), the backbone is frozen (denoted with snowflake), and only the contextual pattern bank is updated through expansion:

$$\mathbf{P}'_\tau = \mathbf{P}_{\tau-1} \parallel \Delta\mathbf{P}_\tau, \tag{4}$$

where $\Delta\mathbf{P}_\tau \in \mathbb{R}^{(N_\tau - N_{\tau-1}) \times d}$ represents newly introduced parameters for the current incremental period. Only the expanded contextual pattern bank $\mathbf{P}'_\tau \in \mathbb{R}^{N_\tau \times d}$ is fine-tuned during training. Notably, even without explicit clustering constraints, the contextual pattern bank autonomously distinguishes heterogeneous and relevant nodes through data-driven parameter learning and prompt-based interactions with the backbone, driven by the prediction task. This strategy ensures that the backbone retains previously acquired knowledge, while the contextual pattern bank continually adapts to evolving distributions. It incrementally expands to represent an increasingly diverse set of environmental patterns, thereby avoiding the inadequacy exhibited by fixed models in novel scenarios.

Distinct from existing work (Wang et al., 2023a; Chen & Liang, 2025; Wang et al., 2023b), we introduce a *Prompt-Based Guidance* (Peebles & Xie, 2023; Zhang et al., 2023) mechanism to enhance $\mathbf{P}_\tau$'s capacity to model both node-level relevance and heterogeneity. Specifically, the contextual pattern bank comprises three groups of trainable parameters: $\mathbf{P}_\tau^{(i)} \in \mathbb{R}^{N_\tau \times d}$ for $i \in 0, 1, 2$. As illustrated in Figure 2, these components interact with the backbone's hidden representation $\mathbf{H}_\tau$ via the following prompt-based gating function:

$$\mathbf{H}'_\tau = \mathbf{P}_\tau^{(1)} \cdot h_\theta(\mathbf{H}_\tau \cdot (1 + \mathbf{P}_\tau^{(0)})), \tag{5}$$

where $h_\theta$ denotes an arbitrary submodule within the backbone. This gating mechanism enables adaptive modeling of node heterogeneity. Additionally, $\mathbf{P}_\tau^{(2)}$ acts as a key embedding in the attention module, guiding the backbone to generalize correlation-aware information under task constraints. Importantly, since the contextual pattern bank encodes high-level abstractions rather than raw historical data, our method supports knowledge retention without revisiting prior data—offering advantages in *privacy protection* and *storage efficiency*.

## 4.3 GENERAL SPATIO-TEMPORAL BACKBONE

While the contextual pattern bank mitigates catastrophic forgetting in continual learning, it lacks the ability to model dynamic spatio-temporal correlations and handle distributional drift. To address this, we design a **general spatio-temporal backbone** aimed at handling distributional drift, spatio-temporal correlation modeling, and graph scalability during continual learning. The term *general* implies that the backbone is independent of the number of nodes and does not rely on any predefined adjacency matrix, making it adaptable to arbitrary spatio-temporal data structures.

As shown in Figure 2, the backbone operates as follows: the input spatio-temporal data first passes through a **frequency-domain network** (`FreNet`), which maps it into high-dimensional temporal representations and extracts stable components via frequency domain analysis. A **dual-stream linear graph attention** (`DLGA`) module then captures dynamic spatial correlations, followed by a feedforward layer with a multilayer perceptron for enhanced nonlinear expressivity. Finally, the features are reconstructed to their original shape by another `FreNet` and passed through a prediction layer. We detail the `FreNet` and `DLGA` modules below.

**Frequency-Domain Network.** Spatio-temporal data in evolving environments often suffer from distributional drift (Wang et al., 2024; Ji et al., 2025; Zhou et al., 2023). Although the contextual pattern bank helps retain stable knowledge, we further address this issue through a dedicated frequency-domain analysis (Xia et al., 2023). `FreNet` is designed to capture temporal correlations while emphasizing stable components in the data, such as periodicity and trends, which are more resilient to distributional changes (Liu & Zhang, 2025). Specifically, `STBP` employs two `FreNet`s—one at the beginning and one at the end of the backbone (Figure 2). The first maps input data $\mathbf{X}_\tau \in \mathbb{R}^{N_\tau \times T_h}$ through a linear layer into a high-dimensional representation $\mathbf{H}_\tau \in \mathbb{R}^{N_\tau \times d}$, which is then transformed to the frequency domain using a Fast Fourier Transform (FFT). A learnable frequency-domain embedding $\mathbf{F}_\tau \in \mathbb{C}^{(\frac{d}{2}+1)}$ adaptively highlights stable features. This process is formalized as:

$$\mathbf{H}_\tau^f = \text{IFFT}(\text{FFT}(\mathbf{H}_\tau) \odot \mathbf{F}_\tau), \tag{6}$$

where $\mathbf{H}_\tau^f \in \mathbb{R}^{N_\tau \times d}$ is further processed by a linear layer. The resulting representation $\mathbf{H}_\tau^f$ then interacts with the contextual pattern bank component $\mathbf{P}_\tau^{(0)}$ via gating-based prompt guidance (Eq. 5) to produce $\mathbf{H}_\tau^s \in \mathbb{R}^{N_\tau \times d}$, which serves as input to the subsequent `DLGA` module. The second `FreNet` performs an inverse operation, restoring the feature shape to $\mathbb{R}^{N_\tau \times T_h}$. Compared to traditional temporal modules like RNNs (Li et al., 2018; Bai et al., 2020) or TCNs (Zheng et al., 2023; Fang et al., 2023), `FreNet` offers higher computational efficiency and enhanced ability to extract

stable low-frequency components (e.g., periodicity and trends) while suppressing high-frequency noise, thereby obtaining more robust temporal representations that are resilient to distributional drift across periods and scenarios.

**Dual-Stream Linear Graph Attention.** After obtaining stable components, it remains essential to capture complex spatial interactions and time-varying node correlations. An effective spatial module must adaptively learn node correlations in a data-driven manner, maintain computational efficiency, and scale to growing graphs. Graph attention mechanisms (Veličković et al., 2018) have emerged as promising solutions, enabling dynamic correlation modeling without relying on fixed adjacency matrices. However, conventional graph attention (Zheng et al., 2020; Jiang et al., 2023a; Liu et al., 2023a) incurs $O(N^2)$ complexity, limiting its scalability. To overcome this, we propose `DLGA` (Figure 2), which improves efficiency using a *random feature mapping*-based linear attention mechanism (Katharopoulos et al., 2020). Moreover, `DLGA` introduces a **dual-stream structure** by incorporating the contextual pattern bank $\mathbf{P}_\tau^{(2)} \in \mathbb{R}^{N_\tau \times d}$ as an additional *key*. This enables the model to assess the relationship between evolving input patterns and stored knowledge. Formally:

$$\mathbf{Q} = \mathbf{W}_q \mathbf{H}_\tau^s, \quad \mathbf{K} = \mathbf{W}_k \mathbf{H}_\tau^s, \quad \mathbf{V} = \mathbf{W}_v \mathbf{H}_\tau^s, \tag{7}$$

$$\begin{aligned}
\mathbf{H}_\tau^{s'} &= \text{Attention}(\mathbf{Q}, \mathbf{K}, \mathbf{V}, \mathbf{P}_\tau^{(2)}) \\
&= \text{Softmax}(\mathbf{Q}\mathbf{K}^\top + \mathbf{Q}(\mathbf{P}_\tau^{(2)})^\top)\mathbf{V},
\end{aligned} \tag{8}$$

$$\begin{aligned}
\text{Attention}(\mathbf{Q}, \mathbf{K}, \mathbf{V}, \mathbf{P}_\tau^{(2)}) &\approx (\phi(\mathbf{Q})\phi(\mathbf{K})^\top + \phi(\mathbf{Q})\phi(\mathbf{P}_\tau^{(2)})^\top)\mathbf{V} \\
&= \phi(\mathbf{Q})\left(\phi(\mathbf{K})^\top \mathbf{V} + \phi(\mathbf{P}_\tau^{(2)})^\top \mathbf{V}\right).
\end{aligned} \tag{9}$$

Here, $\mathbf{W}_q$, $\mathbf{W}_k$, and $\mathbf{W}_v$ are trainable projection matrices. $\mathbf{H}_\tau^s$ and $\mathbf{H}_\tau^{s'} \in \mathbb{R}^{N_\tau \times d}$ denote the input and the spatially enriched representation passed to the feedforward layer of the `DLGA` module, respectively. The function $\phi(\cdot)$ denotes a random feature mapping, with Softmax used for approximation in our implementation. For further details on the approximation derivation, see Appendix A.3.1. Notably, the linear attention approximation does not explicitly construct an adjacency matrix. Instead, it implicitly models dynamic correlations by reordering operations in the attention computation. `DLGA` reduces computational complexity from quadratic to linear, while preserving dynamic spatial modeling and seamlessly integrating prompt-based knowledge from the contextual pattern bank.

## 5 EXPERIMENT

### 5.1 EXPERIMENTAL SETTINGS

**Datasets.** We evaluate our model on three real-world streaming spatio-temporal datasets from the traffic and meteorology domains. The traffic datasets, **PEMS-Stream** (Chen et al., 2001) and **CA-Stream** (Liu et al., 2023b), consist of traffic flow measurements provided by the California Department of Transportation (CalTrans), with a sampling interval of 5 minutes. The meteorological dataset, **AIR-Stream** (Chen & Liang, 2025), is derived from urban air quality platform of the Chinese Environmental Monitoring Center, with hourly sampling intervals. To ensure fair evaluation, all datasets are split into training, validation, and test sets using a fixed ratio of 6:2:2. For each prediction task, the model is trained to forecast the next 12 time steps based on the previous 12 observations. Detailed dataset statistics are provided in Appendix A.4.1.

**Baselines and Metrics.** We select representative models from two categories as baselines: ▷ `Conventional spatio-temporal forecasting models`, including lightweight spatio-temporal architectures such as **GWNet** (Wu et al., 2019), **STID** (Shao et al., 2022a), and **iTransformer** (Liu et al., 2024b). These models are adapted specifically for incremental training in our experiments. ▷ `Continual spatio-temporal forecasting models`, including **TrafficStream**, **STKEC** (Wang et al., 2023a), **PECPM** (Wang et al., 2023b), **STRAP** (Zhang et al., 2025), and **EAC** (Chen & Liang, 2025). The performance of all models is evaluated using the following metrics: Mean Absolute Error (**MAE**), Root Mean Squared Error (**RMSE**), and Mean Absolute Percentage Error (**MAPE**). More details on this are included in Appendix A.4.2.

### 5.2 MAIN RESULTS

The main experimental results are summarized in Table 1, which reports the metrics averaged over all incremental periods. We also present the results at specific forecasting horizons (3, 6, and 12 time steps ahead), together with the overall average across horizons. STGNNs, including GWNet and STID,

Table 1: Main experimental results. **Bold**: best, underline: second best.

| Dataset | Metric | Horizon | GWNet | STID | iTransformer | TrafficStream | STKEC | PECPM | STRAP | EAC | STBP |
|---|---|---|---|---|---|---|---|---|---|---|---|
| PEMS-Stream | MAE | 3 | $19.64_{\pm0.12}$ | $24.34_{\pm0.13}$ | $17.63_{\pm0.76}$ | $14.23_{\pm0.09}$ | $14.29_{\pm0.12}$ | $14.26_{\pm0.13}$ | $14.30_{\pm0.11}$ | $13.86_{\pm0.16}$ | **$11.62_{\pm0.09}$** |
| | | 6 | $19.68_{\pm0.19}$ | $25.45_{\pm0.21}$ | $20.82_{\pm0.76}$ | $16.43_{\pm0.03}$ | $16.44_{\pm0.11}$ | $16.35_{\pm0.10}$ | $16.34_{\pm0.10}$ | $15.40_{\pm0.19}$ | **$12.26_{\pm0.10}$** |
| | | 12 | $20.63_{\pm0.09}$ | $29.42_{\pm0.38}$ | $28.33_{\pm0.86}$ | $21.76_{\pm0.07}$ | $21.66_{\pm0.11}$ | $21.46_{\pm0.19}$ | $21.52_{\pm0.15}$ | $18.90_{\pm0.28}$ | **$13.47_{\pm0.08}$** |
| | | Avg. | $19.87_{\pm0.10}$ | $26.07_{\pm0.23}$ | $21.60_{\pm0.79}$ | $16.95_{\pm0.03}$ | $16.96_{\pm0.09}$ | $16.86_{\pm0.12}$ | $16.88_{\pm0.10}$ | $15.67_{\pm0.20}$ | **$12.31_{\pm0.07}$** |
| | RMSE | 3 | $32.20_{\pm0.17}$ | $39.37_{\pm0.13}$ | $28.20_{\pm1.15}$ | $23.00_{\pm0.09}$ | $23.08_{\pm0.14}$ | $23.07_{\pm0.15}$ | $23.06_{\pm0.13}$ | $22.26_{\pm0.23}$ | **$19.20_{\pm0.13}$** |
| | | 6 | $32.34_{\pm0.32}$ | $40.86_{\pm0.19}$ | $33.80_{\pm1.13}$ | $26.87_{\pm0.04}$ | $26.93_{\pm0.15}$ | $26.76_{\pm0.20}$ | $26.71_{\pm0.14}$ | $24.99_{\pm0.28}$ | **$20.51_{\pm0.15}$** |
| | | 12 | $33.73_{\pm0.09}$ | $46.20_{\pm0.43}$ | $45.98_{\pm1.25}$ | $35.29_{\pm0.11}$ | $35.19_{\pm0.11}$ | $34.77_{\pm0.37}$ | $34.80_{\pm0.19}$ | $30.56_{\pm0.45}$ | **$22.67_{\pm0.13}$** |
| | | Avg. | $32.59_{\pm0.18}$ | $41.67_{\pm0.21}$ | $34.88_{\pm1.17}$ | $27.52_{\pm0.05}$ | $27.56_{\pm0.11}$ | $27.37_{\pm0.20}$ | $27.35_{\pm0.13}$ | $25.30_{\pm0.29}$ | **$20.52_{\pm0.11}$** |
| | MAPE (%) | 3 | $27.47_{\pm0.69}$ | $37.79_{\pm2.23}$ | $32.46_{\pm3.04}$ | $18.34_{\pm0.67}$ | $18.54_{\pm0.61}$ | $18.19_{\pm0.66}$ | $18.69_{\pm0.52}$ | $18.35_{\pm0.31}$ | **$15.00_{\pm0.24}$** |
| | | 6 | $27.22_{\pm0.58}$ | $39.70_{\pm2.43}$ | $36.73_{\pm3.84}$ | $20.77_{\pm0.71}$ | $20.64_{\pm0.48}$ | $20.79_{\pm0.57}$ | $21.33_{\pm0.41}$ | $20.11_{\pm0.36}$ | **$15.55_{\pm0.26}$** |
| | | 12 | $29.38_{\pm1.18}$ | $47.94_{\pm2.91}$ | $54.31_{\pm4.66}$ | $27.88_{\pm0.26}$ | $27.05_{\pm0.62}$ | $28.33_{\pm0.52}$ | $28.20_{\pm1.10}$ | $24.30_{\pm0.57}$ | **$16.75_{\pm0.23}$** |
| | | Avg. | $27.79_{\pm0.76}$ | $41.09_{\pm2.49}$ | $39.63_{\pm3.81}$ | $21.66_{\pm0.54}$ | $21.50_{\pm0.45}$ | $21.73_{\pm0.45}$ | $22.17_{\pm0.46}$ | $20.42_{\pm0.41}$ | **$15.65_{\pm0.21}$** |
| CA-Stream | MAE | 3 | $23.49_{\pm0.80}$ | $27.71_{\pm0.23}$ | $20.16_{\pm0.06}$ | $17.82_{\pm0.26}$ | $17.69_{\pm0.19}$ | $17.93_{\pm0.12}$ | $23.59_{\pm0.61}$ | $17.66_{\pm0.37}$ | **$15.01_{\pm0.18}$** |
| | | 6 | $23.31_{\pm0.69}$ | $28.93_{\pm0.26}$ | $24.37_{\pm0.06}$ | $20.38_{\pm0.17}$ | $20.41_{\pm0.04}$ | $20.33_{\pm0.09}$ | $25.38_{\pm0.68}$ | $19.68_{\pm0.54}$ | **$15.78_{\pm0.07}$** |
| | | 12 | $24.78_{\pm0.86}$ | $33.61_{\pm0.45}$ | $34.05_{\pm0.06}$ | $26.92_{\pm0.53}$ | $27.05_{\pm0.17}$ | $26.68_{\pm0.19}$ | $31.10_{\pm0.89}$ | $24.86_{\pm1.33}$ | **$17.19_{\pm0.09}$** |
| | | Avg. | $23.73_{\pm0.75}$ | $29.71_{\pm0.28}$ | $25.34_{\pm0.05}$ | $21.09_{\pm0.29}$ | $21.09_{\pm0.13}$ | $21.04_{\pm0.11}$ | $26.25_{\pm0.62}$ | $20.20_{\pm0.69}$ | **$15.77_{\pm0.09}$** |
| | RMSE | 3 | $35.87_{\pm0.98}$ | $41.53_{\pm0.31}$ | $31.58_{\pm0.09}$ | $28.01_{\pm0.22}$ | $28.02_{\pm0.19}$ | $28.00_{\pm0.16}$ | $34.73_{\pm0.74}$ | $27.46_{\pm0.46}$ | **$24.37_{\pm0.27}$** |
| | | 6 | $35.68_{\pm0.88}$ | $43.14_{\pm0.35}$ | $37.76_{\pm0.10}$ | $32.19_{\pm0.22}$ | $32.43_{\pm0.05}$ | $31.94_{\pm0.09}$ | $37.97_{\pm0.86}$ | $30.64_{\pm0.83}$ | **$25.71_{\pm0.22}$** |
| | | 12 | $37.57_{\pm1.11}$ | $49.18_{\pm0.58}$ | $51.24_{\pm0.10}$ | $41.59_{\pm0.64}$ | $42.08_{\pm0.21}$ | $41.14_{\pm0.30}$ | $46.74_{\pm1.36}$ | $37.77_{\pm1.94}$ | **$28.08_{\pm0.14}$** |
| | | Avg. | $36.20_{\pm0.96}$ | $44.12_{\pm0.37}$ | $38.94_{\pm0.09}$ | $33.01_{\pm0.35}$ | $33.24_{\pm0.13}$ | $32.77_{\pm0.17}$ | $39.05_{\pm0.80}$ | $31.18_{\pm0.99}$ | **$25.70_{\pm0.16}$** |
| | MAPE (%) | 3 | $24.61_{\pm0.95}$ | $29.24_{\pm0.65}$ | $21.76_{\pm0.17}$ | $17.05_{\pm0.41}$ | $16.60_{\pm0.19}$ | $17.63_{\pm0.91}$ | $19.11_{\pm0.49}$ | $18.26_{\pm1.88}$ | **$14.22_{\pm0.03}$** |
| | | 6 | $24.44_{\pm0.80}$ | $30.66_{\pm0.78}$ | $26.76_{\pm0.22}$ | $19.22_{\pm0.30}$ | $18.98_{\pm0.17}$ | $19.74_{\pm0.92}$ | $20.48_{\pm0.39}$ | $19.45_{\pm1.16}$ | **$14.85_{\pm0.07}$** |
| | | 12 | $25.71_{\pm0.46}$ | $36.88_{\pm1.29}$ | $39.81_{\pm0.38}$ | $25.47_{\pm0.46}$ | $24.99_{\pm0.29}$ | $25.94_{\pm1.19}$ | $24.97_{\pm0.59}$ | $24.52_{\pm1.10}$ | **$16.20_{\pm0.08}$** |
| | | Avg. | $24.79_{\pm0.85}$ | $31.73_{\pm0.86}$ | $28.34_{\pm0.20}$ | $19.98_{\pm0.30}$ | $19.61_{\pm0.19}$ | $20.49_{\pm0.91}$ | $21.15_{\pm0.47}$ | $20.17_{\pm1.25}$ | **$14.94_{\pm0.05}$** |
| AIR-Stream | MAE | 3 | $28.48_{\pm1.43}$ | $32.85_{\pm0.21}$ | $22.37_{\pm0.76}$ | $20.73_{\pm0.40}$ | $20.95_{\pm0.17}$ | $20.82_{\pm0.35}$ | $21.41_{\pm0.33}$ | $20.41_{\pm0.36}$ | **$20.00_{\pm0.14}$** |
| | | 6 | $29.79_{\pm0.89}$ | $33.15_{\pm0.22}$ | $26.22_{\pm0.48}$ | $25.64_{\pm0.34}$ | $25.54_{\pm0.08}$ | $25.54_{\pm0.19}$ | $26.12_{\pm0.34}$ | $25.20_{\pm0.29}$ | **$24.70_{\pm0.30}$** |
| | | 12 | $31.30_{\pm0.52}$ | $33.88_{\pm0.25}$ | $29.45_{\pm0.31}$ | $29.04_{\pm0.23}$ | $28.94_{\pm0.12}$ | $28.95_{\pm0.11}$ | $29.38_{\pm0.31}$ | $28.57_{\pm0.42}$ | **$28.28_{\pm0.63}$** |
| | | Avg. | $29.66_{\pm1.01}$ | $33.23_{\pm0.22}$ | $25.53_{\pm0.56}$ | $24.58_{\pm0.24}$ | $24.63_{\pm0.11}$ | $24.60_{\pm0.21}$ | $25.16_{\pm0.32}$ | $24.21_{\pm0.43}$ | **$23.64_{\pm0.23}$** |
| | RMSE | 3 | $44.38_{\pm2.04}$ | $51.24_{\pm0.28}$ | $34.98_{\pm1.18}$ | $32.80_{\pm0.57}$ | $33.13_{\pm0.28}$ | $33.07_{\pm0.52}$ | $33.72_{\pm0.41}$ | $32.19_{\pm0.57}$ | **$32.15_{\pm0.24}$** |
| | | 6 | $46.22_{\pm1.28}$ | $51.61_{\pm0.31}$ | $40.95_{\pm0.73}$ | $40.41_{\pm0.53}$ | $40.38_{\pm0.20}$ | $40.48_{\pm0.39}$ | $41.13_{\pm0.40}$ | **$39.63_{\pm0.43}$** | $39.81_{\pm0.26}$ |
| | | 12 | $48.34_{\pm0.85}$ | $52.55_{\pm0.39}$ | $45.70_{\pm0.55}$ | $45.54_{\pm0.47}$ | $45.53_{\pm0.27}$ | $45.63_{\pm0.27}$ | $46.07_{\pm0.34}$ | **$44.65_{\pm0.63}$** | $44.97_{\pm0.97}$ |
| | | Avg. | $46.01_{\pm1.46}$ | $51.72_{\pm0.33}$ | $39.67_{\pm0.91}$ | $38.58_{\pm0.53}$ | $38.70_{\pm0.26}$ | $38.76_{\pm0.41}$ | $39.37_{\pm0.38}$ | $37.83_{\pm0.60}$ | **$37.76_{\pm0.30}$** |
| | MAPE (%) | 3 | $38.02_{\pm2.60}$ | $43.52_{\pm0.64}$ | $28.64_{\pm1.28}$ | $26.33_{\pm0.30}$ | $26.24_{\pm0.30}$ | $25.79_{\pm0.50}$ | $26.80_{\pm0.36}$ | $26.06_{\pm0.71}$ | **$24.64_{\pm0.16}$** |
| | | 6 | $39.98_{\pm1.70}$ | $44.12_{\pm0.57}$ | $34.91_{\pm0.68}$ | $33.33_{\pm0.21}$ | $33.10_{\pm0.28}$ | $32.97_{\pm0.18}$ | $33.30_{\pm0.19}$ | $32.88_{\pm0.64}$ | **$30.66_{\pm0.42}$** |
| | | 12 | $42.37_{\pm1.14}$ | $45.06_{\pm0.62}$ | $40.79_{\pm0.39}$ | $39.27_{\pm0.24}$ | $39.02_{\pm0.18}$ | $38.67_{\pm0.02}$ | $38.87_{\pm0.24}$ | $38.85_{\pm0.67}$ | **$36.23_{\pm0.52}$** |
| | | Avg. | $39.87_{\pm1.87}$ | $44.16_{\pm0.60}$ | $34.15_{\pm0.76}$ | $32.29_{\pm0.29}$ | $32.12_{\pm0.21}$ | $31.82_{\pm0.19}$ | $32.37_{\pm0.28}$ | $31.77_{\pm0.53}$ | **$29.70_{\pm0.35}$** |

rely on static graph assumptions and are not designed for continual learning. Following prior work (Chen & Liang, 2025), we therefore **retrain** the backbone from scratch at each incremental stage using only data from the current period. In contrast, iTransformer is scenario-agnostic, so we adopt an **online** training regime: at each stage it is trained on the complete node set of the current spatio-temporal graph,

Table 2: Comparison of few-shot forecasting performance.

| Model | PEMS-Stream 10% | | | CA-Stream 10% | | |
|---|---|---|---|---|---|---|
| | MAE | RMSE | MAPE (%) | MAE | RMSE | MAPE (%) |
| GWNet | $30.15_{\pm1.06}$ | $45.30_{\pm1.59}$ | $48.80_{\pm3.85}$ | $33.73_{\pm0.89}$ | $50.80_{\pm1.43}$ | $36.52_{\pm0.86}$ |
| STID | $33.42_{\pm2.90}$ | $50.63_{\pm3.73}$ | $63.96_{\pm12.60}$ | $37.09_{\pm0.52}$ | $55.10_{\pm0.69}$ | $39.18_{\pm1.12}$ |
| iTransformer | $20.99_{\pm0.19}$ | $32.67_{\pm0.25}$ | $49.11_{\pm1.62}$ | $25.43_{\pm0.08}$ | $39.01_{\pm0.10}$ | $28.39_{\pm0.54}$ |
| TrafficStream | $17.23_{\pm0.08}$ | $27.49_{\pm0.17}$ | $27.63_{\pm0.43}$ | $21.28_{\pm0.19}$ | $33.25_{\pm0.22}$ | $20.45_{\pm0.45}$ |
| STKEC | $17.75_{\pm0.12}$ | $28.23_{\pm0.13}$ | $27.80_{\pm0.88}$ | $21.20_{\pm0.13}$ | $33.20_{\pm0.08}$ | $20.23_{\pm0.46}$ |
| PECPM | $17.05_{\pm0.02}$ | $27.20_{\pm0.07}$ | $29.08_{\pm1.90}$ | $21.48_{\pm0.15}$ | $33.33_{\pm0.13}$ | $21.25_{\pm0.86}$ |
| STRAP | $17.68_{\pm0.10}$ | $27.98_{\pm0.14}$ | $31.67_{\pm2.88}$ | $26.34_{\pm0.79}$ | $39.39_{\pm1.09}$ | $21.34_{\pm0.45}$ |
| EAC | $16.13_{\pm0.05}$ | $25.57_{\pm0.06}$ | $24.02_{\pm1.23}$ | $20.94_{\pm0.70}$ | $32.15_{\pm1.00}$ | $21.37_{\pm1.53}$ |
| STBP | **$13.58_{\pm0.05}$** | **$22.24_{\pm0.13}$** | **$17.89_{\pm0.29}$** | **$17.11_{\pm0.03}$** | **$27.48_{\pm0.16}$** | **$17.60_{\pm0.30}$** |

initialized from the previous period's weights, enabling end-to-end fine-tuning. More detailed experimental results are provided in Appendix A.4.4.

**Results of conventional methods.** As shown in Table 1, STGNNs trained from scratch achieve only poor performance on all datasets. Although these methods work well under static assumptions, they fail to exploit past spatio-temporal knowledge, resulting in unsatisfactory performance. In contrast, iTransformer performs better by leveraging historical spatio-temporal information through online training, but it still suffers from catastrophic forgetting and is therefore not an ideal solution.

**Results of CSTF methods.** The best-performing models are those that explicitly mitigate catastrophic forgetting, including CSTF methods such as PECPM, STRAP, and EAC. Compared with full-parameter fine-tuning strategies (e.g., PECPM, STKEC, TrafficStream), lightweight prompt-based adaptation on a frozen backbone (e.g., EAC, STRAP, STBP) yields higher average accuracy, highlighting the benefits of dynamically tuning only a small set of parameters. Nevertheless, STRAP performs notably poorly on CA-Stream, indicating that retrieval-based pattern matching struggles

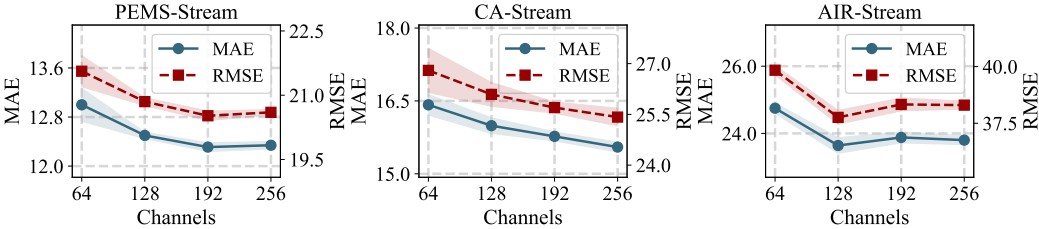

Figure 4: Results of ablation experiments.

Figure 5: Results of parameter experiments.

in extreme incremental scenarios with rapid, large-scale topology expansion. Overall, our proposed `STBP` outperforms all competing models. Compared with the best baseline, `STBP` reduces the average MAE by **21.44%**, **21.93%**, and **2.35%** on the PEMS-Stream, CA-Stream, and AIR-Stream datasets, respectively. This gain stems from the bridge it establishes between STGNNs and CSTF methods: the carefully designed general spatio-temporal backbone and contextual pattern bank jointly capture dynamic spatio-temporal correlations, thereby mitigating catastrophic forgetting and alleviating distributional drift.

**Results of few-shot forecasting task.** To further evaluate the robustness of the proposed model under low-resource scenarios, we construct a few-shot training setting and compare it against existing baselines. Specifically, we simulate a few-shot setting in which the sample size of the first incremental period is kept unchanged, while the training set size for subsequent periods is reduced to only 10% of the original. The test set size remains fixed throughout. As shown in Table 2, `STBP` consistently outperforms all other methods, highlighting its strong ability to extract meaningful patterns from limited data. CSTF baselines are more resilient to low-resource conditions than conventional STGNNs (e.g., GWNet, STID). This demonstrates that when data is extremely scarce, conventional models struggle to capture stable spatio-temporal patterns, whereas CSTF methods can leverage knowledge accumulated from historical stages to adapt more quickly to new nodes. The continual learning mechanism effectively mitigates catastrophic forgetting, allowing the model to continuously utilize previously learned general features during incremental learning.

### 5.3 ABLATION STUDY & PARAMETER SENSITIVITY ANALYSIS

**Ablation Study Settings.** To validate the core contributions of `STBP`, we design the following variants for ablation experiments: ❶ **Retrain**: The contextual pattern bank is removed. Similar to GWNet and STID, a new backbone is trained for each incremental period using the spatio-temporal graph data of that period, with the corresponding model predicting the results for the current test set. ❷ **Online**: The contextual pattern bank is removed. Similar to iTransformer, the model is trained on the complete node data of the current spatio-temporal graph and initialized with the model from the previous period, allowing for adjustments across the entire model. ❸ **w/o Backbone**: The contextual pattern bank is retained, but the spatio-temporal backbone is replaced with the ones used in TrafficStream, STKEC, and EAC—i.e., replacing `FreNet` and `DLGA` with CNN and GCN. ❹ **w/o DLGA**: The `DLGA` module in the spatio-temporal backbone is ablated. ❺ **EAC**: We also include EAC, which follows a similar approach, for comparison in the ablation study.

**Ablation findings.** As shown in Figure 4, the ablation results demonstrate that parameter expansion in the contextual pattern bank, together with spatio-temporal pattern distinction and prompt guidance, is essential for alleviating catastrophic forgetting in continual learning. The performance of the **Retrain** and **Online** variants supports this conclusion. Notably, even without the contextual pattern

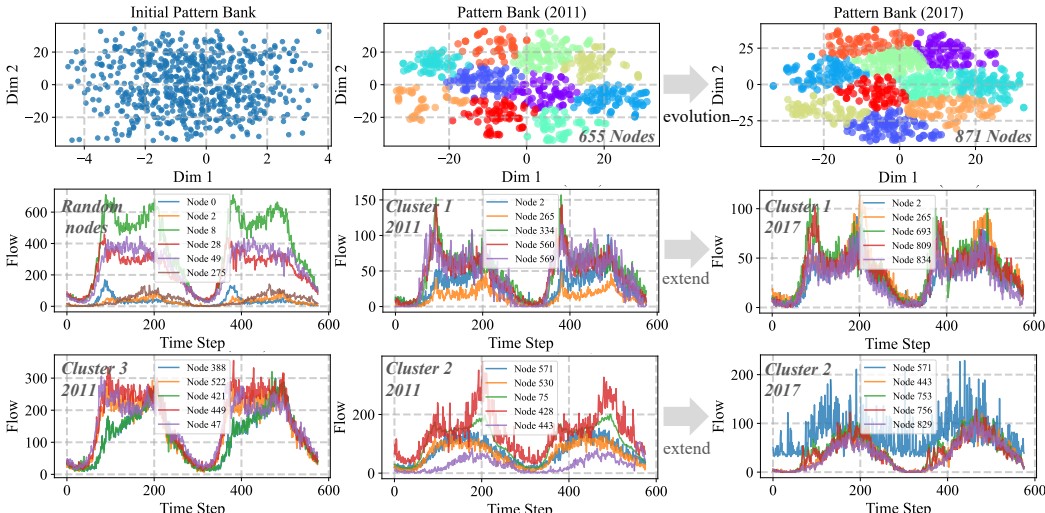

Figure 6: Case study on PEMS-Stream.

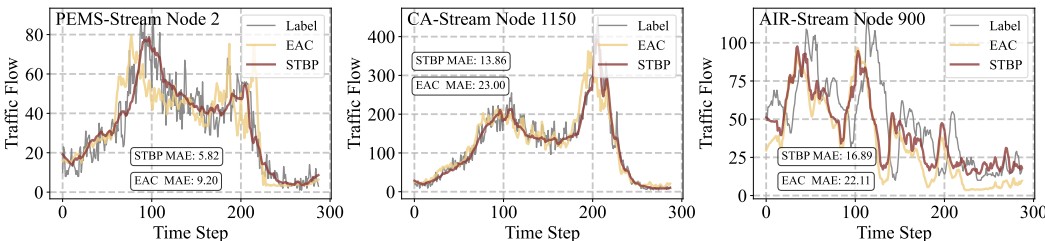

Figure 7: Visualization of real forecasting results.

bank on traffic datasets, the spatio-temporal backbone alone attains performance comparable to **EAC** under online training, highlighting the critical role of real-time dynamic correlation modeling and temporal distribution-drift mitigation in adapting to new incremental tasks. The performance drop observed in the **w/o Backbone** variant further confirms the indispensability of the general backbone and highlights the portability and adaptability of the pattern bank across different backbone architectures. Moreover, removing the `DLGA` module leads to significant performance degradation, validating its role in capturing dynamic spatial correlations and integrating prompt-based knowledge. The `FreNet` module also makes a notable contribution by improving computational efficiency and enhancing the extraction of stable temporal components.

**Parameter Sensitivity Analysis.** Additionally, we perform a sensitivity analysis on the adjustable hyperparameter $d$ in `STBP`. In `STBP`, $d$ represents the feature dimension for each module's feature mapping, as well as the feature dimension of parameters in the contextual pattern bank. The analysis results are shown in Figure 5. Increasing $d$ enhances the model's overall parameter count and improves its expressive power. However, the performance gains from increasing $d$ do not grow indefinitely; after reaching a certain threshold, the performance gain stabilizes. Further increases in $d$ not only fail to improve performance but may also lead to negative effects, causing parameter redundancy. More parameter sensitivity analysis can be found in Appendix A.4.5.

## 5.4 CASE STUDY

To illustrate the distinction and expandability of the contextual pattern bank in `STBP`, we apply t-SNE to reduce the dimensionality of $\mathbf{P}_\tau \in \mathbb{R}^{N_\tau \times d}$ on the PEMS-Stream dataset. As shown in Figure 6, each point represents a graph node. Initially untrained, the pattern bank shows a chaotic distribution. After incremental training, clear clusters emerge. Nodes within the same cluster exhibit similar periodic and trend patterns in their traffic data, while those in different clusters (e.g., Clusters 1–3) show distinct behaviors. New nodes from later stages (e.g., Nodes 693, 809, 834 in 2017) are

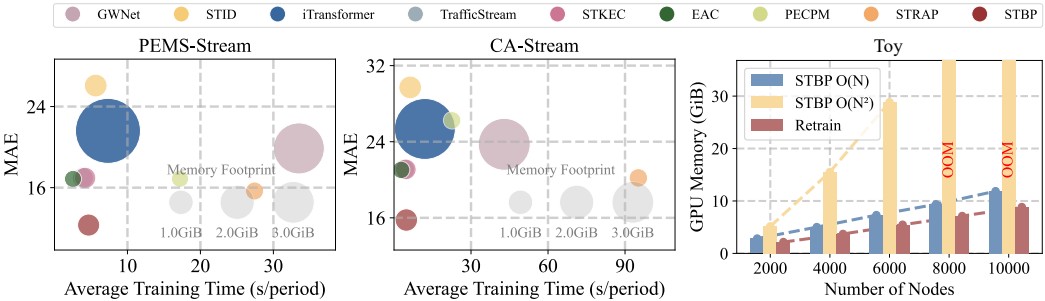

Figure 8: Efficiency comparison. STBP $O(N^2)$ denotes the version without linear attention, and Retrain refers to removing the contextual pattern bank.

correctly grouped into existing clusters, demonstrating that the pattern bank effectively distinguishes and generalizes spatio-temporal patterns through parameter fine-tuning, enabling continual adaptation.

In addition, we conduct an intuitive comparison of the forecasting performance between STBP and EAC in real-world application scenarios. As shown in Figure 7, we select representative nodes from three datasets for visualization. Compared to EAC, STBP more accurately captures dynamic trends, and its predictions demonstrate higher practical relevance in real-world continual learning environments. Additional case studies on other datasets can be found in Appendix A.4.6.

## 5.5 EFFICIENCY STUDY

An effective CSTF method must balance scalability, computational cost, and performance. We evaluate the efficiency of STBP against baselines under the same settings. As shown in Figure 8, the average computational cost per period on PEMS-Stream and AIR-Stream is reported, with scatter size indicating GPU memory usage. We further analyze the impact of linear attention, full attention, and removal of the contextual pattern bank using a toy dataset. Results indicate that non-continual methods—such as GWNet, STID, and iTransformer—require global parameter adjustments at each phase, impairing efficiency. iTransformer, in particular, incurs high memory overhead due to quadratic attention complexity. Even lightweight non-continual models exhibit limited efficiency in incremental training.

In contrast, CSTF methods such as EAC, TrafficStream, and STKEC achieve higher efficiency through lightweight backbones and localized parameter tuning. While PECPM and STRAP maintain low memory usage, their training speeds remain modest. Despite its more complex backbone, STBP incurs only minimal overhead compared to models like EAC, thanks to optimizations including frequency-domain processing and linear attention. This enables STBP to deliver substantial performance gains with negligible cost increase. Results on the toy dataset confirm that linear attention reduces computational load effectively. As node count grows, the contextual pattern bank introduces only linear additional cost through its lightweight interaction with the backbone, avoiding exponential overhead. Furthermore, on CA-Stream, STBP maintains state-of-the-art performance even under drastic graph expansion, demonstrating strong scalability.

## 6 CONCLUSION

In this work, we propose STBP, a novel framework for continual spatio-temporal forecasting. By combining a general-purpose backbone with a scalable contextual pattern bank, STBP efficiently mitigates catastrophic forgetting while capturing dynamic spatio-temporal correlations. It adapts to evolving urban data without retraining from scratch, making it suitable for real-time applications. Validated on multiple datasets, STBP demonstrates strong continual learning capabilities. Nevertheless, STBP currently supports continual learning in a single-task setting. In the future, we plan to extend its application to cross-domain continual spatio-temporal forecasting, which will be a crucial step towards developing a foundational spatio-temporal model.

## ACKNOWLEDGMENTS

This work was partly supported by the National Key Research and Development Program of China under Grant 2022YFB4501704, the National Natural Science Foundation of China under Grant 72342026, and Fundamental Research Funds for the Central Universities under Grant 2024-6-ZD-02.

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

# A  APPENDIX

## A.1  NOTATIONS

Table 3 summarizes the notations frequently used throughout this manuscript.

Table 3: The notations that are commonly used in the manuscript.

| Notation | Definition |
|---|---|
| $\mathbb{G} = \{G_\tau\}_{\tau=1}^{\mathcal{T}}$ | Streaming spatio-temporal graph |
| $\mathbf{X}_\tau$ | Inputs for the $\tau$ period |
| $\mathbf{Y}_\tau$ | Prediction for the $\tau$ period |
| $\mathbf{P}_\tau$ | contextual pattern bank for the $\tau$ period |
| $\mathbf{P}'_\tau$ | Expanded contextual pattern bank |
| $\mathbf{H}_\tau$ | Hidden representation for the $\tau$ period |
| $\mathcal{M}_\theta$ | Spatio-Temporal backbone |
| $\mathbf{F}_\tau$ | Frequency domain embedding |
| $\mathbf{H}_\tau^f$ | Representation after frequency-domain processing |
| $\mathbf{W}_q$ | Trainable parameter weights |
| $\mathbf{W}_k$ | Trainable parameter weights |
| $\mathbf{W}_v$ | Trainable parameter weights |
| $\phi(\cdot)$ | Random mapping function |
| $\mathbf{H}_\tau^s$ | Input of the `DLGA` module |
| $\mathbf{H}_\tau^{s'}$ | Output of the `DLGA` module |

## A.2  RELATED WORK DETAILS

### A.2.1  SPATIO-TEMPORAL FORECASTING

Spatio-temporal forecasting aims to support decision-making in critical domains such as intelligent transportation and smart cities by uncovering dynamic correlations embedded in spatio-temporal data. These data typically exhibit strong spatial-temporal correlations and pronounced heterogeneity. In recent years, deep learning-based STGNNs have emerged as effective tools for such forecasting tasks. STGNNs generally employ temporal modules (e.g., recurrent neural networks (RNNs) (Li et al., 2018; Jiang et al., 2023b; Shao et al., 2022b) and convolutional neural networks (CNNs)) (Yu et al., 2018; Liu & Zhang, 2024b;a) to capture temporal correlations, while leveraging spatial modules (e.g., graph neural networks (GNNs)) (Veličković et al., 2018; Song et al., 2020) to model spatial relationships.

Early STGNNs, such as STGCN (Yu et al., 2018) and DCRNN (Li et al., 2018), combined basic temporal and spatial components for forecasting tasks, often relying on predefined geographic adjacency matrices. However, these static assumptions hinder their ability to model dynamically changing spatial correlations in a data-driven manner. Subsequent works—such as GWNet (Wu et al., 2019), DGCRN (Li et al., 2023), and MegaCRN (Jiang et al., 2023b)—introduced adaptive adjacency matrices or learned spatial correlations directly from data, significantly improving prediction accuracy. More recent advances, including STID (Shao et al., 2022a), STAEformer (Liu et al., 2023a), and HimNet (Dong et al., 2024), have highlighted the importance of spatial pattern distinction in enhancing forecasting performance. These models incorporate trainable mechanisms such as spatial embeddings, parameter pools, and contextual pattern banks to distinguish spatial patterns more precisely, thereby improving both accuracy and adaptability.

Despite these advancements, most existing STGNNs are built on static assumptions and are not designed to operate in dynamic, continually evolving spatio-temporal environments—limiting their applicability in continual learning scenarios.

### A.2.2 CONTINUAL SPATIO-TEMPORAL LEARNING

Early research in continual learning primarily focused on computer vision (Lee & Park, 2024; Miao et al., 2024a) and natural language processing (Caccia et al., 2020; Xia et al., 2025). With the rapid development of IoT and intelligent transportation systems, attention has increasingly shifted toward CSTF (Chen et al., 2021; Wang et al., 2023a; Chen & Liang, 2025; Wang et al., 2023b; Miao et al., 2025), which addresses the challenges of dynamically evolving and expanding spatio-temporal data. CSTF aims to enable models to continuously learn and adapt to new patterns and knowledge in changing environments, while minimizing forgetting of previously acquired information or performance degradation.

One of the earliest frameworks in this domain, TrafficStream (Chen et al., 2021), pioneered the integration of spatio-temporal modeling with continual learning. It employed strategies such as historical data replay and parameter smoothing to handle long-term streaming traffic data, achieving accurate traffic flow forecasting. Subsequently, the STKEC (Wang et al., 2023a) introduced an influence-based knowledge expansion strategy and a memory-augmented knowledge consolidation mechanism to better accommodate the growth of transportation networks while mitigating catastrophic forgetting. The PECPM (Wang et al., 2023b) framework employs a pattern matching mechanism to maintain and dynamically update a bank of representative traffic patterns from evolving road networks, enabling efficient continual learning without historical data and improving both prediction accuracy and training efficiency. Meanwhile, STRAP (Zhang et al., 2025) introduces a retrieval-augmented approach that builds multi-dimensional pattern libraries. During inference, it retrieves and fuses relevant historical patterns with current inputs via a plug-and-play prompting mechanism, effectively boosting generalization in OOD scenarios while mitigating catastrophic forgetting.

The EAC (Chen & Liang, 2025) further advanced the field by incorporating prompt tuning, enabling CSTF with a small number of trainable parameters. Its dynamic prompt pool, which supports both "expansion" and "compression," enhances adaptability to new nodes while preserving historical knowledge, improving both generalization and computational efficiency. In addition, the UFCL (Miao et al., 2025) leveraged federated learning to preserve data privacy and introduced a global replay buffer for synthetic spatio-temporal data, addressing the challenges of distributed streaming environments. Despite these advancements, most existing methods primarily focus on alleviating knowledge forgetting, while overlooking the critical role of the spatio-temporal backbone in continual learning scenarios.

**Advancements Beyond Existing Prompt Methods.** Unlike EAC's "expand-and-compress" prompt pool that may lead to historical information loss during compression, our contextual pattern bank adopts pure parametric incremental expansion without compression, more completely preserving historical knowledge. Additionally, while EAC's prompt interaction is relatively simple (e.g., feature addition), our pattern bank employs structured multi-component design ($\mathbf{P}_\tau^{(0)}, \mathbf{P}_\tau^{(1)}, \mathbf{P}_\tau^{(2)}$) that jointly models node relevance and heterogeneity through gating and attention mechanisms, enabling more comprehensive spatio-temporal representation learning.

### A.3 FURTHER METHODS DETAILS

### A.3.1 APPROXIMATION DERIVATION OF EQ. 9

An approximate derivation of the attention mechanism in the dual-stream linear graph attention is presented below:

$$
\begin{aligned}
\text{Attention}\left(\boldsymbol{q}_u, \boldsymbol{k}_v, \boldsymbol{v}_v, \boldsymbol{p}_v\right) &= \sum_{v=1}^{N} \frac{\exp\left(\boldsymbol{q}_u^\top \boldsymbol{k}_v\right) \boldsymbol{v}_v}{\sum_{w=1}^{N} \exp\left(\boldsymbol{q}_u^\top \boldsymbol{k}_w\right)} + \sum_{v=1}^{N} \frac{\exp\left(\boldsymbol{q}_u^\top \boldsymbol{p}_v\right) \boldsymbol{v}_v}{\sum_{w=1}^{N} \exp\left(\boldsymbol{q}_u^\top \boldsymbol{p}_w\right)} \\
&\approx \frac{\sum_{v=1}^{N} \phi\left(\boldsymbol{q}_u\right)^\top \phi\left(\boldsymbol{k}_v\right) \boldsymbol{v}_v}{\sum_{w=1}^{N} \phi\left(\boldsymbol{q}_u\right)^\top \phi\left(\boldsymbol{k}_w\right)} + \frac{\sum_{v=1}^{N} \phi\left(\boldsymbol{q}_u\right)^\top \phi\left(\boldsymbol{p}_v\right) \boldsymbol{v}_v}{\sum_{w=1}^{N} \phi\left(\boldsymbol{q}_u\right)^\top \phi\left(\boldsymbol{p}_w\right)} \\
&= \underbrace{\left[\frac{\phi\left(\boldsymbol{q}_u\right)^\top \sum_{v=1}^{N} \phi\left(\boldsymbol{k}_v\right) \boldsymbol{v}_v^\top}{\phi\left(\boldsymbol{q}_u\right)^\top \sum_{w=1}^{N} \phi\left(\boldsymbol{k}_w\right)}\right]}_{\text{Term 1: Representation-based aggregation}} + \underbrace{\left[\frac{\phi\left(\boldsymbol{q}_u\right)^\top \sum_{v=1}^{N} \phi\left(\boldsymbol{p}_v\right) \boldsymbol{v}_v^\top}{\phi\left(\boldsymbol{q}_u\right)^\top \sum_{w=1}^{N} \phi\left(\boldsymbol{p}_w\right)}\right]}_{\text{Term 2: Prompt-based aggregation}}
\end{aligned}
\tag{10}
$$

---

**Algorithm 1** The workflow of STBP for continual spatio-temporal forecasting

---

**Require:** Spatio-temporal backbone $\mathcal{M}_\theta$, contextual pattern bank $\{\mathbf{P}_1, \mathbf{P}_2, \ldots, \mathbf{P}_\tau\}$, streaming train data $\{\mathbf{X}_1, \mathbf{X}_2 \ldots, \mathbf{X}_\tau\}$.
**Ensure:** Optimized backbone $\mathcal{M}_{\theta*}$ and contextual pattern bank $\{\mathbf{P}_1^*, \ldots, \mathbf{P}_\tau^*\}$
    **Initialize:**$\mathcal{M}_\theta \leftarrow \{\}, \mathbf{P}_1 \leftarrow \{\}$
    **for** each period $i$ in $\{1, 2, 3, \ldots, \tau\}$ **do**
        **if** $i == 1$ **then**
            ▷ *Initial training phase*                                           ◁
            Construct the initial contextual pattern bank $\mathbf{P}_1$
            Optimize backbone and contextual pattern bank with initial data $\mathbf{X}_1$:
            $(\mathcal{M}_{\theta*}, \mathbf{P}_1^*) \leftarrow \operatorname{argmin}_\theta \mathcal{M}_\theta(\mathbf{X}_1, \mathbf{P}_1)$
        **else**
            ▷ *Streaming learning phase*                                     ◁
            Expand contextual pattern bank $\mathbf{P}_i$: $\mathbf{P}_i \leftarrow \mathbf{P}_{i-1} \parallel \Delta\mathbf{P}_i$
            Inherit parameters: $(\mathcal{M}_\theta, \mathbf{P}_i) \leftarrow (\mathcal{M}_{\theta*}, \mathbf{P}_{i-1}^*)$
            Freeze backbone parameters $\mathcal{M}_\theta$: $\theta \leftarrow \mathtt{freeze}(\theta)$
            Fine-tune $\mathbf{P}_i$ with backbone $\mathcal{M}_\theta$ on $\mathbf{X}_i$:
            $\mathbf{P}_i^* \leftarrow \operatorname{argmin}_\theta \mathcal{M}_\theta(\mathbf{X}_i, \mathbf{P}_i)$

---

Table 4: Overview of continual spatio-temporal forecasting datasets.

| Dataset | Domain | Time Range | Period | Node Expansion | Frequency |
|---------|--------|------------|--------|----------------|-----------|
| **PEMS-Stream** | Traffic | 07/10/2011 - 09/08/2017 | 7 | $655 \rightarrow 715 \rightarrow 786$ $\rightarrow 822 \rightarrow 834 \rightarrow 850$ $\rightarrow 871$ | 5 min |
| **CA-Stream** | Traffic | 01/01/2019 - 04/30/2019 | 4 | $480 \rightarrow 691 \rightarrow 1175$ $\rightarrow 1698$ | 5 min |
| **AIR-Stream** | Air Quality | 01/01/2016 - 12/31/2019 | 4 | $1087 \rightarrow 1154$ $\rightarrow 1193 \rightarrow 1202$ | 1 hour |

where $\boldsymbol{q}_u$ is the query tensor of node $u$; $\boldsymbol{k}_v$ and $\boldsymbol{v}_v$ are the key and value tensors of node $v$, respectively; and $\boldsymbol{p}_v$ represents the prompt information for node $v$.

### A.3.2 ALGORITHM WORKFLOW

The overall workflow of STBP for continual spatio-temporal forecasting is presented in a more intuitive manner in Algorithm 1.

### A.4 ADDITIONAL EXPERIMENT DETAILS

### A.4.1 DATASET DETAILS

Table 4 and Table 5 jointly summarize the characteristics of the three continual spatio-temporal datasets used in this study: **PEMS-Stream**, **CA-Stream**, and **AIR-Stream**. These datasets differ in domain (traffic [1] vs. air quality [2]), temporal span, and topological evolution, collectively covering a broad spectrum of real-world non-stationary scenarios suitable for evaluating continual learning models. PEMS-Stream contains highway traffic sensor readings collected across California from July 2011 to September 2017. It spans seven periods with a gradual increase in the number of sensor nodes—from 655 to 871—resulting in a +33% relative growth. This dataset simulates realistic, long-term infrastructure expansion and serves as a benchmark for evaluating model adaptability under progressive and stable topological changes. CA-Stream, also in the traffic domain, covers a much

---

[1]https://dot.ca.gov/programs/traffic-operations/mpr/pems-source
[2]https://air.cnemc.cn:18007/

Table 5: Topological dynamics and evaluation purposes of the datasets.

| Dataset | Topology Change | ∆Nodes | Relative Change | Primary Purpose |
|---|---|---|---|---|
| PEMS-Stream | Gradual expansion | 216 | +33% | Realistic progressive growth |
| CA-Stream | Explosive expansion | 1,218 | +254% | Extreme incremental stress test |
| AIR-Stream | Stable expansion | 115 | +10% | Cross-domain validation |

shorter period (January to April 2019) but features a sharp and sudden node expansion—from 480 to 1,698—corresponding to a +254% relative increase. This explosive growth intro-

Table 6: Distribution shift analysis based on MMD tests.

| Type | PEMS-Stream | | AIR-Stream | | CA-Stream | |
|---|---|---|---|---|---|---|
| | MMD | $p$ | MMD | $p$ | MMD | $p$ |
| Original Node | 0.0939 | 0.008 | 0.3324 | 0.001 | 0.0792 | 0.1119 |
| Added Node | 0.2958 | 0.001 | 0.2679 | 0.001 | 0.3361 | 0.0010 |

duces significant distributional shifts, making CA-Stream a challenging testbed for assessing model robustness under rapidly evolving conditions. AIR-Stream focuses on urban air quality and environmental measurements from 2016 to 2019. It exhibits modest but steady node growth—from 1,087 to 1,202 (+10%)—and represents a relatively stable expansion setting. Its distinct domain and smoother structural changes make it particularly suitable for evaluating cross-domain generalization and robustness to gradual environmental variation.

To further assess non-stationarity, we conduct Maximum Mean Discrepancy (MMD) tests across different periods, separately evaluating *original nodes* (present from the beginning) and *added nodes* (introduced during expansion), as shown in Table 6. A distribution shift is considered significant when MMD $> 0.1$ or $p < 0.05$. Across all datasets, added nodes consistently exhibit stronger distributional shifts, reflecting the spatial disruptions caused by topological expansion. For instance, CA-Stream shows a substantial shift for added nodes (MMD = 0.3361, $p = 0.0010$), consistent with its rapid growth. Interestingly, AIR-Stream records the highest MMD among original nodes (0.3324, $p = 0.001$), despite minimal structural change—indicating notable temporal drift in environmental data. This highlights AIR-Stream's importance for evaluating robustness to evolving distributions even under stable topology. By contrast, PEMS-Stream shows only moderate drift among original nodes (MMD = 0.0939), aligning with its smoother expansion. CA-Stream presents weaker drift in original nodes (MMD = 0.0792, $p = 0.1119$), likely due to its limited temporal span. These results underscore the dual challenge in continual spatio-temporal learning: managing both spatial shifts induced by node expansion and temporal non-stationarity inherent to dynamic environments, with their nature and intensity varying across domains.

### A.4.2 BASELINES AND METRICS DETAILS

In this paper, we provide a detailed comparison with two categories of representative models:

**Conventional Spatio-Temporal Forecasting Models.** ❶ **GWNet** (Wu et al., 2019): A STGNN model based on an adaptive adjacency matrix that can adaptively capture latent spatial dependencies. This model combines graph convolutional networks and temporal convolutions to effectively capture spatio-temporal correlations in the data. ❷ **STID** (Shao et al., 2022a): An efficient multilayer perceptron model that solves the problem of sample non-separability using trainable embeddings, showing outstanding performance in spatio-temporal forecasting tasks. ❸ **iTransformer** (Liu et al., 2024b): A time-series model that does not rely on a static graph structure. By modeling the interactions between variables, it captures temporal features and is effectively applied to multivariate time series forecasting tasks.

**Continual Spatio-Temporal Forecasting Models.** These models are designed to handle time-varying data and are suitable for continual training tasks. Like `STBP`, they belong to the category of continual learning models. We selected the following three representative models for comparison: ❶ **TrafficStream** (Chen et al., 2021): The first model for CSTF, it employs a traffic pattern fusion approach, historical data replay, and parameter smoothing strategies to efficiently integrate and learn new spatio-temporal patterns in the continuously expanding and evolving traffic network. ❷ **STKEC** (Wang et al., 2023a): A traffic forecasting model based on the continual learning paradigm. Through an influence-based knowledge expansion strategy and a memory-augmented knowledge

Table 7: Comparison of prediction performance for each incremental period on PEMS-Stream. **Bold**: best, underline: second best.

| Model | Metric | PEMS-Stream Period | | | | | | | |
|---|---|---|---|---|---|---|---|---|---|
| | | 2011 | 2012 | 2013 | 2014 | 2015 | 2016 | 2017 | Avg. |
| GWNet | MAE | $25.90_{\pm1.19}$ | $20.51_{\pm2.14}$ | $18.59_{\pm1.31}$ | $16.17_{\pm0.39}$ | $19.37_{\pm1.73}$ | $17.58_{\pm0.83}$ | $20.95_{\pm1.03}$ | $19.87_{\pm0.10}$ |
| | RMSE | $44.79_{\pm1.08}$ | $32.43_{\pm2.60}$ | $29.32_{\pm1.47}$ | $25.37_{\pm0.60}$ | $31.37_{\pm2.11}$ | $29.89_{\pm1.14}$ | $34.96_{\pm1.12}$ | $32.59_{\pm0.18}$ |
| | MAPE (%) | $29.79_{\pm1.61}$ | $29.18_{\pm2.95}$ | $28.89_{\pm2.63}$ | $25.45_{\pm2.51}$ | $29.61_{\pm4.28}$ | $24.79_{\pm1.44}$ | $26.86_{\pm2.42}$ | $27.79_{\pm0.76}$ |
| STID | MAE | $32.68_{\pm1.34}$ | $26.10_{\pm0.97}$ | $25.36_{\pm2.28}$ | $22.78_{\pm1.25}$ | $23.37_{\pm1.38}$ | $24.28_{\pm1.09}$ | $27.98_{\pm1.08}$ | $26.07_{\pm0.23}$ |
| | RMSE | $53.22_{\pm2.16}$ | $41.02_{\pm1.64}$ | $39.50_{\pm3.50}$ | $35.67_{\pm2.07}$ | $37.15_{\pm2.34}$ | $40.94_{\pm1.78}$ | $44.20_{\pm1.39}$ | $41.67_{\pm0.21}$ |
| | MAPE (%) | $44.33_{\pm5.66}$ | $38.68_{\pm1.13}$ | $41.40_{\pm0.58}$ | $39.50_{\pm3.34}$ | $42.05_{\pm5.09}$ | $39.65_{\pm6.86}$ | $42.05_{\pm4.57}$ | $41.09_{\pm2.49}$ |
| iTransformer | MAE | $25.44_{\pm3.24}$ | $20.90_{\pm0.70}$ | $20.37_{\pm0.58}$ | $20.58_{\pm0.85}$ | $20.23_{\pm0.71}$ | $20.83_{\pm0.80}$ | $22.83_{\pm1.04}$ | $21.60_{\pm0.79}$ |
| | RMSE | $39.73_{\pm4.15}$ | $33.32_{\pm1.23}$ | $32.39_{\pm1.00}$ | $33.25_{\pm1.35}$ | $33.04_{\pm1.04}$ | $35.70_{\pm1.22}$ | $36.69_{\pm1.64}$ | $34.88_{\pm1.17}$ |
| | MAPE (%) | $35.71_{\pm2.97}$ | $38.61_{\pm4.15}$ | $37.35_{\pm3.82}$ | $42.89_{\pm5.17}$ | $40.49_{\pm6.04}$ | $40.83_{\pm5.24}$ | $41.55_{\pm4.25}$ | $39.63_{\pm3.81}$ |
| TrafficStream | MAE | $18.15_{\pm0.19}$ | $16.81_{\pm0.36}$ | $16.16_{\pm0.12}$ | $16.62_{\pm0.14}$ | $16.39_{\pm0.01}$ | $16.47_{\pm0.15}$ | $18.04_{\pm0.09}$ | $16.95_{\pm0.03}$ |
| | RMSE | $27.75_{\pm0.23}$ | $26.72_{\pm0.58}$ | $25.64_{\pm0.13}$ | $26.96_{\pm0.06}$ | $27.29_{\pm0.02}$ | $28.93_{\pm0.05}$ | $29.31_{\pm0.10}$ | $27.52_{\pm0.05}$ |
| | MAPE (%) | $21.35_{\pm0.76}$ | $21.14_{\pm0.46}$ | $21.33_{\pm0.85}$ | $22.61_{\pm1.31}$ | $21.36_{\pm0.53}$ | $20.84_{\pm0.86}$ | $22.99_{\pm1.47}$ | $21.66_{\pm0.54}$ |
| STKEC | MAE | $18.09_{\pm0.46}$ | $16.83_{\pm0.36}$ | $16.26_{\pm0.20}$ | $16.48_{\pm0.24}$ | $16.38_{\pm0.15}$ | $16.31_{\pm0.13}$ | $18.41_{\pm0.35}$ | $16.96_{\pm0.09}$ |
| | RMSE | $27.47_{\pm0.47}$ | $26.85_{\pm0.56}$ | $25.74_{\pm0.21}$ | $26.97_{\pm0.40}$ | $27.17_{\pm0.22}$ | $28.70_{\pm0.38}$ | $30.03_{\pm0.68}$ | $27.56_{\pm0.11}$ |
| | MAPE (%) | $21.00_{\pm0.77}$ | $21.42_{\pm1.07}$ | $20.54_{\pm0.47}$ | $21.53_{\pm0.63}$ | $21.71_{\pm0.46}$ | $20.38_{\pm0.86}$ | $23.87_{\pm1.01}$ | $21.50_{\pm0.52}$ |
| PECPM | MAE | $18.43_{\pm0.41}$ | $16.91_{\pm0.43}$ | $16.03_{\pm0.28}$ | $16.27_{\pm0.12}$ | $16.09_{\pm0.13}$ | $16.21_{\pm0.23}$ | $18.05_{\pm0.39}$ | $16.86_{\pm0.12}$ |
| | RMSE | $28.09_{\pm0.39}$ | $26.94_{\pm0.64}$ | $25.48_{\pm0.43}$ | $26.49_{\pm0.28}$ | $26.71_{\pm0.28}$ | $28.55_{\pm0.24}$ | $29.31_{\pm0.63}$ | $27.37_{\pm0.20}$ |
| | MAPE (%) | $21.57_{\pm0.84}$ | $21.18_{\pm0.63}$ | $20.71_{\pm0.82}$ | $22.82_{\pm1.61}$ | $22.12_{\pm1.68}$ | $20.99_{\pm0.37}$ | $22.73_{\pm1.83}$ | $21.73_{\pm0.45}$ |
| STRAP | MAE | $18.18_{\pm0.08}$ | $17.40_{\pm0.56}$ | $16.07_{\pm0.11}$ | $16.30_{\pm0.06}$ | $16.04_{\pm0.06}$ | $16.16_{\pm0.13}$ | $18.02_{\pm0.12}$ | $16.88_{\pm0.10}$ |
| | RMSE | $27.72_{\pm0.04}$ | $27.60_{\pm0.82}$ | $25.46_{\pm0.16}$ | $26.49_{\pm0.06}$ | $26.53_{\pm0.07}$ | $28.34_{\pm0.18}$ | $29.30_{\pm0.19}$ | $27.35_{\pm0.13}$ |
| | MAPE (%) | $22.66_{\pm0.84}$ | $21.06_{\pm0.61}$ | $22.58_{\pm0.79}$ | $22.87_{\pm0.51}$ | $21.57_{\pm1.66}$ | $21.90_{\pm0.73}$ | $22.54_{\pm1.14}$ | $22.17_{\pm0.46}$ |
| EAC | MAE | $18.12_{\pm0.26}$ | $15.41_{\pm0.40}$ | $14.67_{\pm0.21}$ | $15.09_{\pm0.15}$ | $14.96_{\pm0.15}$ | $14.78_{\pm0.21}$ | $16.67_{\pm0.19}$ | $15.67_{\pm0.20}$ |
| | RMSE | $27.72_{\pm0.70}$ | $24.23_{\pm0.62}$ | $23.08_{\pm0.32}$ | $24.29_{\pm0.22}$ | $24.58_{\pm0.25}$ | $26.26_{\pm0.30}$ | $26.96_{\pm0.30}$ | $25.30_{\pm0.29}$ |
| | MAPE (%) | $19.62_{\pm0.16}$ | $19.90_{\pm0.51}$ | $20.40_{\pm0.63}$ | $20.01_{\pm0.32}$ | $20.72_{\pm0.27}$ | $19.98_{\pm0.70}$ | $22.33_{\pm1.19}$ | $20.42_{\pm0.41}$ |
| STBP | MAE | $\mathbf{14.29_{\pm0.05}}$ | $\mathbf{12.13_{\pm0.11}}$ | $\mathbf{11.60_{\pm0.09}}$ | $\mathbf{11.71_{\pm0.07}}$ | $\mathbf{11.61_{\pm0.06}}$ | $\mathbf{11.38_{\pm0.09}}$ | $\mathbf{13.42_{\pm0.17}}$ | $\mathbf{12.31_{\pm0.07}}$ |
| | RMSE | $\mathbf{22.00_{\pm0.07}}$ | $\mathbf{19.47_{\pm0.18}}$ | $\mathbf{18.58_{\pm0.15}}$ | $\mathbf{19.54_{\pm0.10}}$ | $\mathbf{19.55_{\pm0.07}}$ | $\mathbf{21.91_{\pm0.07}}$ | $\mathbf{22.56_{\pm0.26}}$ | $\mathbf{20.52_{\pm0.11}}$ |
| | MAPE (%) | $\mathbf{16.34_{\pm0.12}}$ | $\mathbf{15.26_{\pm0.28}}$ | $\mathbf{15.38_{\pm0.15}}$ | $\mathbf{15.51_{\pm0.23}}$ | $\mathbf{15.47_{\pm0.35}}$ | $\mathbf{14.80_{\pm0.23}}$ | $\mathbf{16.76_{\pm0.20}}$ | $\mathbf{15.65_{\pm0.21}}$ |

consolidation mechanism, STKEC helps the model effectively integrate new spatio-temporal traffic patterns in an ever-expanding road network while retaining previously learned spatio-temporal patterns. ❸ **PECPM** (Wang et al., 2023b): A continual spatio-temporal forecasting model for evolving traffic networks, relying on a pattern-matching pattern bank to store representative patterns without full historical data. It fine-tunes with new/conflict nodes for knowledge expansion and uses preservation/traceability mechanisms to avoid forgetting. ❹ **STRAP** (Zhang et al., 2025): A retrieval-augmented framework for OOD generalization, building a multi-dimensional key-value pattern library (spatial/temporal/spatio-temporal) during training. It retrieves similar patterns to fuse with current data in inference, achieving SOTA without task-specific fine-tuning. ❺ **EAC** (Chen & Liang, 2025): A CSTF based on prompt tuning. By integrating a base STGNN with a continual prompt pool, it efficiently addresses incremental learning and catastrophic forgetting in streaming data using lightweight trainable parameters.

**The Excluded Models.** Some baselines that might be considered relevant for comparison were excluded, and we provide explanations for their exclusion below. ❶ **STAEformer** (Liu et al., 2023a): a widely recognized baseline, was not included in our comparison due to non-convergence observed when applying the same experimental setting as used for GWNet and STID on the selected three datasets. To ensure fair and unambiguous evaluation, we excluded it from the results and have provided the corresponding training logs in the code repository. ❷ **UFCL** (Miao et al., 2025): The CSTF method UFCL is not included in the comparison due to differences in experimental settings, which prevent a fair evaluation.

**Metrics Details.** Additionally, the performance metrics used in the experiments to evaluate the model, namely MAE, RMSE, and MAPE, are defined as follows:

$$\text{MAE} = \frac{1}{n} \sum_{i=1}^{n} |y_i - \hat{y}_i| \tag{11}$$

Table 8: Comparison of prediction performance for each incremental period on CA-Stream and AIR-Stream. **Bold**: best, underline: second best.

| Model | Metric | CA-Stream Period | | | | | AIR-Stream Period | | | | |
|---|---|---|---|---|---|---|---|---|---|---|---|
| | | Jan-19 | Feb-19 | Mar-19 | Apr-19 | Avg. | 2016 | 2017 | 2018 | 2019 | Avg. |
| **GWNet** | MAE | $26.94_{\pm1.39}$ | $26.20_{\pm1.56}$ | $20.98_{\pm0.48}$ | $20.82_{\pm0.24}$ | $23.73_{\pm0.75}$ | $35.54_{\pm3.70}$ | $29.94_{\pm1.11}$ | $26.87_{\pm1.57}$ | $26.29_{\pm1.96}$ | $29.66_{\pm1.01}$ |
| | RMSE | $41.01_{\pm1.71}$ | $39.79_{\pm1.97}$ | $32.44_{\pm0.55}$ | $31.58_{\pm0.13}$ | $36.20_{\pm0.96}$ | $55.84_{\pm5.75}$ | $45.28_{\pm1.33}$ | $43.63_{\pm2.52}$ | $39.29_{\pm2.53}$ | $46.01_{\pm1.46}$ |
| | MAPE (%) | $23.13_{\pm0.91}$ | $30.31_{\pm2.77}$ | $23.72_{\pm1.09}$ | $22.00_{\pm0.38}$ | $24.79_{\pm0.85}$ | $41.77_{\pm4.84}$ | $34.94_{\pm0.62}$ | $41.09_{\pm1.99}$ | $41.68_{\pm3.65}$ | $39.87_{\pm1.87}$ |
| **STID** | MAE | $33.83_{\pm0.51}$ | $30.83_{\pm0.23}$ | $26.66_{\pm0.46}$ | $27.50_{\pm0.73}$ | $29.71_{\pm0.28}$ | $40.38_{\pm0.56}$ | $34.92_{\pm0.82}$ | $28.36_{\pm0.43}$ | $29.28_{\pm0.16}$ | $33.23_{\pm0.22}$ |
| | RMSE | $49.98_{\pm0.53}$ | $45.98_{\pm0.25}$ | $39.84_{\pm0.51}$ | $40.70_{\pm0.97}$ | $44.12_{\pm0.37}$ | $64.10_{\pm0.91}$ | $52.12_{\pm1.11}$ | $46.78_{\pm0.78}$ | $43.90_{\pm0.42}$ | $51.72_{\pm0.33}$ |
| | MAPE (%) | $31.21_{\pm2.21}$ | $33.63_{\pm0.89}$ | $31.11_{\pm2.31}$ | $30.95_{\pm1.25}$ | $31.73_{\pm0.86}$ | $44.78_{\pm0.79}$ | $42.04_{\pm0.82}$ | $45.92_{\pm0.63}$ | $43.87_{\pm1.16}$ | $44.16_{\pm0.60}$ |
| **iTransformer** | MAE | $30.00_{\pm0.11}$ | $25.99_{\pm0.08}$ | $23.27_{\pm0.08}$ | $22.11_{\pm0.06}$ | $25.34_{\pm0.05}$ | $32.03_{\pm1.25}$ | $27.07_{\pm0.35}$ | $20.74_{\pm0.18}$ | $22.28_{\pm0.68}$ | $25.53_{\pm0.56}$ |
| | RMSE | $45.21_{\pm0.25}$ | $40.76_{\pm0.15}$ | $35.89_{\pm0.10}$ | $33.90_{\pm0.18}$ | $38.94_{\pm0.09}$ | $50.92_{\pm2.20}$ | $40.22_{\pm0.61}$ | $34.16_{\pm0.39}$ | $33.37_{\pm0.65}$ | $39.67_{\pm0.91}$ |
| | MAPE (%) | $31.23_{\pm0.90}$ | $29.31_{\pm0.59}$ | $28.31_{\pm0.70}$ | $24.50_{\pm0.92}$ | $28.34_{\pm0.20}$ | $38.05_{\pm2.33}$ | $32.25_{\pm0.33}$ | $33.64_{\pm0.43}$ | $32.66_{\pm0.69}$ | $34.15_{\pm0.76}$ |
| **TrafficStream** | MAE | $23.63_{\pm0.42}$ | $21.87_{\pm0.28}$ | $19.71_{\pm0.37}$ | $19.15_{\pm0.18}$ | $21.09_{\pm0.29}$ | **$30.09_{\pm0.63}$** | $26.43_{\pm0.33}$ | $20.34_{\pm0.39}$ | $21.48_{\pm0.31}$ | $24.58_{\pm0.34}$ |
| | RMSE | $36.46_{\pm0.53}$ | $34.85_{\pm0.34}$ | $31.07_{\pm0.43}$ | $29.64_{\pm0.20}$ | $33.01_{\pm0.35}$ | **$48.04_{\pm1.10}$** | $39.24_{\pm0.40}$ | $34.22_{\pm0.68}$ | $32.80_{\pm0.42}$ | $38.58_{\pm0.53}$ |
| | MAPE (%) | $21.34_{\pm1.65}$ | $20.03_{\pm0.36}$ | $19.65_{\pm0.81}$ | $18.88_{\pm0.33}$ | $19.98_{\pm0.30}$ | $34.34_{\pm0.33}$ | $31.34_{\pm0.31}$ | $33.17_{\pm1.08}$ | $30.30_{\pm0.38}$ | $32.29_{\pm0.29}$ |
| **STKEC** | MAE | $23.22_{\pm0.54}$ | $22.29_{\pm0.35}$ | $19.79_{\pm0.26}$ | $19.09_{\pm0.13}$ | $21.09_{\pm0.13}$ | $30.12_{\pm0.30}$ | $26.74_{\pm0.41}$ | $20.34_{\pm0.27}$ | $21.33_{\pm0.31}$ | $24.63_{\pm0.11}$ |
| | RMSE | $36.16_{\pm0.62}$ | $35.73_{\pm0.52}$ | $31.36_{\pm0.38}$ | $29.71_{\pm0.23}$ | $33.24_{\pm0.13}$ | $48.48_{\pm0.61}$ | $39.57_{\pm0.57}$ | $34.33_{\pm0.70}$ | $32.45_{\pm0.29}$ | $38.70_{\pm0.26}$ |
| | MAPE (%) | $20.12_{\pm0.46}$ | $20.01_{\pm0.13}$ | $19.80_{\pm0.79}$ | $18.48_{\pm0.14}$ | $19.61_{\pm0.19}$ | $33.22_{\pm0.32}$ | $31.87_{\pm0.39}$ | $32.71_{\pm0.48}$ | $30.69_{\pm0.35}$ | $32.12_{\pm0.21}$ |
| **PECPM** | MAE | $24.29_{\pm0.22}$ | $21.46_{\pm0.11}$ | $19.34_{\pm0.10}$ | $19.06_{\pm0.15}$ | $21.04_{\pm0.11}$ | $30.86_{\pm1.38}$ | $26.02_{\pm0.60}$ | $20.43_{\pm0.06}$ | $21.15_{\pm0.45}$ | $24.74_{\pm0.25}$ |
| | RMSE | $37.05_{\pm0.42}$ | $34.11_{\pm0.10}$ | $30.54_{\pm0.25}$ | $29.39_{\pm0.17}$ | $32.77_{\pm0.17}$ | $49.63_{\pm2.19}$ | $38.92_{\pm0.60}$ | $34.66_{\pm0.05}$ | $32.28_{\pm0.49}$ | $39.00_{\pm0.48}$ |
| | MAPE (%) | $23.05_{\pm3.21}$ | $20.07_{\pm0.43}$ | $19.77_{\pm1.00}$ | $19.07_{\pm0.21}$ | $20.49_{\pm0.91}$ | $34.29_{\pm0.90}$ | $30.69_{\pm0.63}$ | $31.84_{\pm0.92}$ | $30.22_{\pm0.25}$ | $31.93_{\pm0.22}$ |
| **STRAP** | MAE | $30.64_{\pm1.85}$ | $25.93_{\pm0.48}$ | $23.87_{\pm0.44}$ | $24.56_{\pm0.66}$ | $26.25_{\pm0.62}$ | $32.14_{\pm1.35}$ | $26.36_{\pm0.78}$ | $20.22_{\pm0.66}$ | $21.91_{\pm0.30}$ | $25.16_{\pm0.32}$ |
| | RMSE | $44.86_{\pm2.30}$ | $38.96_{\pm0.60}$ | $35.98_{\pm0.70}$ | $36.41_{\pm0.61}$ | $39.05_{\pm0.80}$ | $51.46_{\pm2.36}$ | $38.99_{\pm0.73}$ | $34.07_{\pm1.51}$ | $32.97_{\pm0.10}$ | $39.37_{\pm0.38}$ |
| | MAPE (%) | $21.72_{\pm0.97}$ | $22.02_{\pm0.63}$ | $20.41_{\pm0.41}$ | $20.43_{\pm0.49}$ | $21.15_{\pm0.47}$ | $34.25_{\pm0.97}$ | $31.62_{\pm0.94}$ | $32.30_{\pm1.13}$ | $31.33_{\pm0.87}$ | $32.37_{\pm0.28}$ |
| **EAC** | MAE | $22.70_{\pm0.82}$ | $20.76_{\pm0.73}$ | $18.77_{\pm0.66}$ | $18.55_{\pm0.57}$ | $20.20_{\pm0.69}$ | $30.36_{\pm1.21}$ | $25.74_{\pm0.33}$ | $19.74_{\pm0.34}$ | $21.02_{\pm0.23}$ | $24.21_{\pm0.43}$ |
| | RMSE | $34.85_{\pm1.19}$ | $32.70_{\pm0.96}$ | $29.04_{\pm0.96}$ | $28.12_{\pm0.87}$ | $31.18_{\pm0.99}$ | $48.33_{\pm1.71}$ | $38.22_{\pm0.24}$ | $32.88_{\pm0.51}$ | **$31.89_{\pm0.26}$** | $37.83_{\pm0.60}$ |
| | MAPE (%) | $20.56_{\pm1.51}$ | $20.72_{\pm1.94}$ | $20.26_{\pm1.18}$ | $19.14_{\pm0.90}$ | $20.17_{\pm1.25}$ | $33.43_{\pm0.72}$ | $30.78_{\pm0.52}$ | $32.02_{\pm1.31}$ | $30.86_{\pm0.31}$ | $31.77_{\pm0.53}$ |
| **STBP** | MAE | **$17.88_{\pm0.37}$** | **$15.86_{\pm0.10}$** | **$14.62_{\pm0.20}$** | **$14.73_{\pm0.22}$** | **$15.77_{\pm0.09}$** | $30.95_{\pm0.55}$ | **$24.32_{\pm0.09}$** | **$18.82_{\pm0.20}$** | **$20.47_{\pm0.15}$** | **$23.64_{\pm0.23}$** |
| | RMSE | **$29.56_{\pm0.84}$** | **$26.54_{\pm0.14}$** | **$23.58_{\pm0.30}$** | **$23.13_{\pm0.34}$** | **$25.70_{\pm0.16}$** | $49.34_{\pm0.79}$ | **$37.65_{\pm0.18}$** | **$32.14_{\pm0.24}$** | $31.92_{\pm0.34}$ | **$37.76_{\pm0.30}$** |
| | MAPE (%) | **$15.47_{\pm0.32}$** | **$14.86_{\pm0.05}$** | **$14.30_{\pm0.12}$** | **$15.11_{\pm0.14}$** | **$14.94_{\pm0.05}$** | **$31.89_{\pm0.48}$** | **$27.98_{\pm0.28}$** | **$30.22_{\pm0.42}$** | **$28.72_{\pm0.32}$** | **$29.70_{\pm0.35}$** |

$$\text{RMSE} = \sqrt{\frac{1}{n}\sum_{i=1}^{n}(y_i - \hat{y}_i)^2} \tag{12}$$

$$\text{MAPE} = \frac{1}{n}\sum_{i=1}^{n}\left|\frac{y_i - \hat{y}_i}{y_i}\right| \times 100\% \tag{13}$$

where $n$ represents the number of observed samples, $y_i$ denotes the $i$-th real sample, and $\hat{y}_i$ is the corresponding predicted value.

### A.4.3 IMPLEMENTATION DETAILS

All experiments are conducted on a machine with an NVIDIA Tesla V100 GPU and 32 GB of memory. The Adam optimizer, with an initial learning rate of 0.01, is used to optimize the training process. The batch size is set to 64, the number of training epochs is set to 200, and an early stopping mechanism is implemented to ensure efficient convergence. The reported results for all baselines are the average of five repeated runs.

### A.4.4 EXPERIMENT RESULTS

Tables 7 and 8 report detailed results for each incremental period, where the metrics of a given period are averaged over 12 forecasting steps. Figure 9 provides a visual summary of the same results to better illustrate the performance trends across periods. Overall, STBP consistently achieves strong performance throughout the entire continual spatio-temporal forecasting process, including both the aggregated performance across all periods and the period-wise results. This advantage is largely attributed to the well-designed spatio-temporal backbone and the contextual pattern bank, which together support effective knowledge reuse and adaptation under evolving spatio-temporal patterns.

### A.4.5 PARAMETER SENSITIVITY ANALYSIS

Beyond the feature dimension $d$, we further investigated the sensitivity of two key architectural hyperparameters: the number of DLGA layers and attention heads. As shown in Figure 10, increasing either parameter yields marginal gains at best, and in some cases, even leads to slight performance

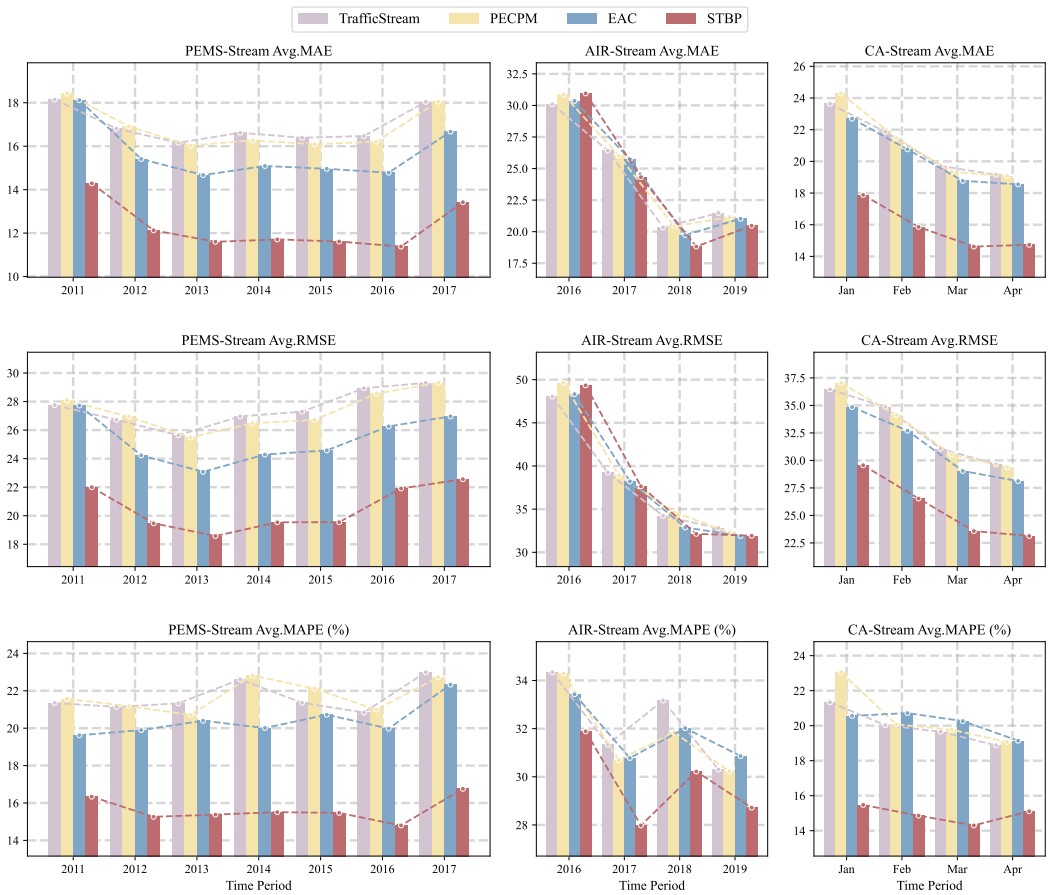

Figure 9: Visualization of period-wise forecasting performance across incremental periods.

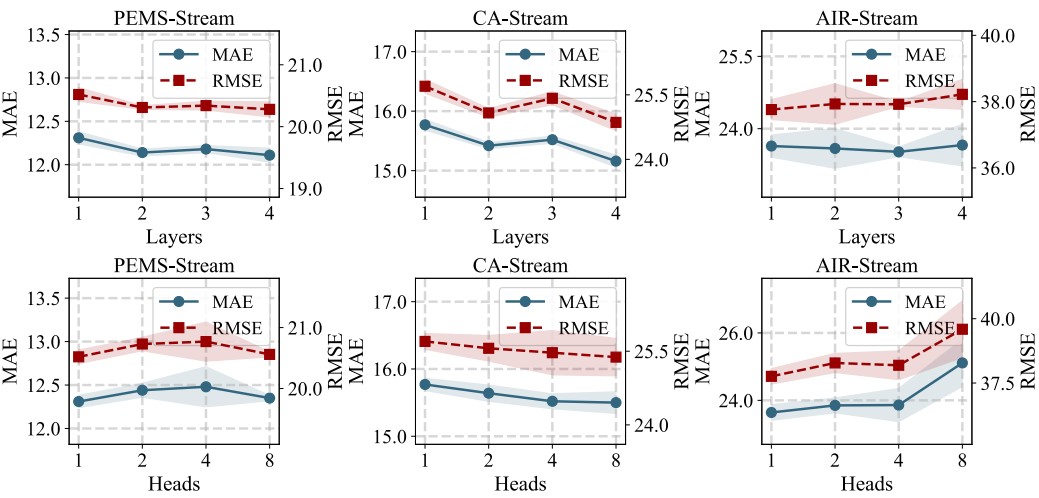

Figure 10: Additional Results of parameter experiments.

degradation. Overall, apart from the feature dimension, model performance remains relatively insensitive to these hyperparameter variations.

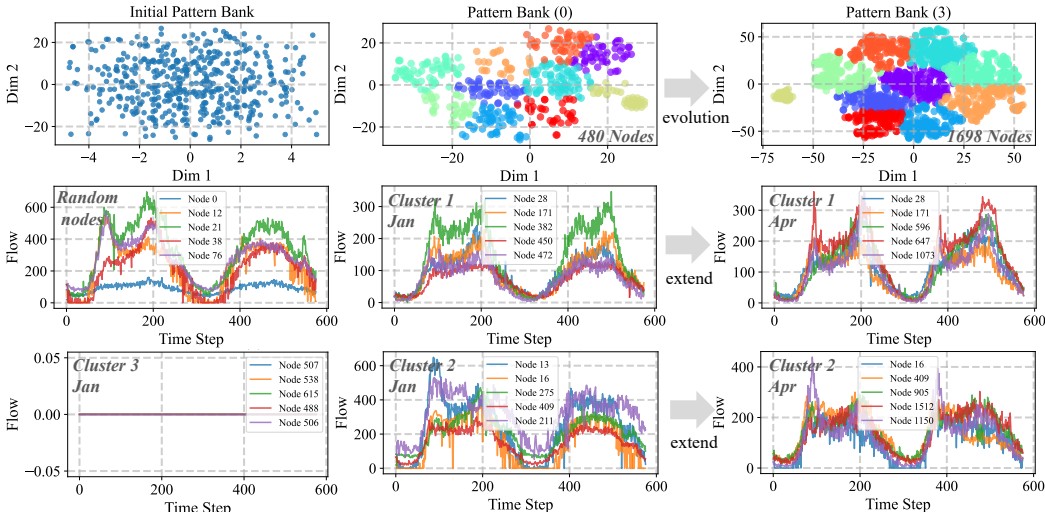

Figure 11: Case Study on CA-Stream.

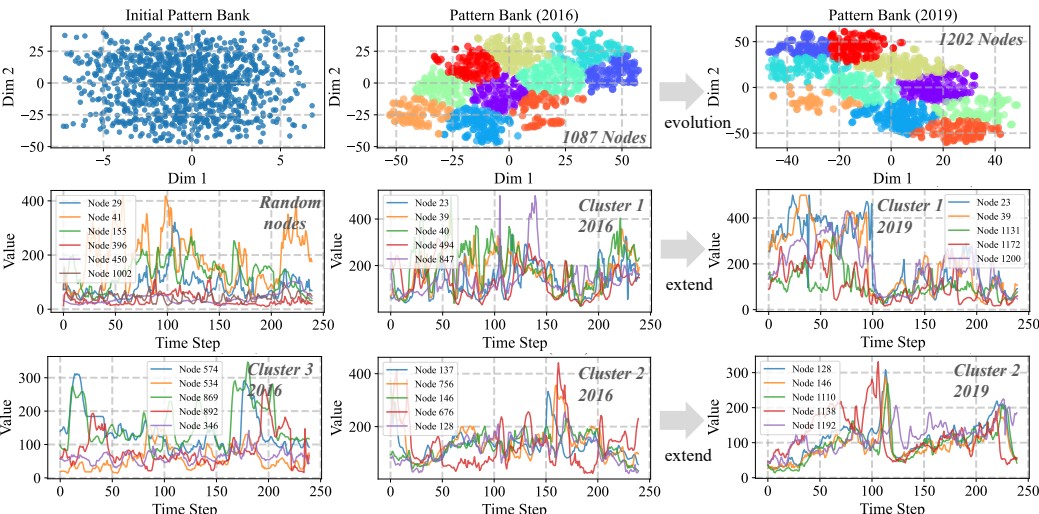

Figure 12: Case Study on AIR-Stream.

### A.4.6 ADDITIONAL CASE STUDY

To maintain consistency with the case study on PEMS-Stream, we also conduct case studies on the CA-Stream and AIR-Stream datasets to further validate the expansion and distinction capabilities of the contextual pattern bank in STBP. The experimental results for CA-Stream are shown in Figure 11. Even in the more challenging task of node increment, STBP's contextual pattern bank effectively distinguishes and consolidates different spatio-temporal patterns, incorporating new patterns introduced by newly added nodes into the existing pattern clusters.

Figure 12 presents the results on AIR-Stream. Compared to traffic flow data, the spatio-temporal patterns in this dataset exhibit more complex periodic and trend changes. Nonetheless, STBP continues to accurately differentiate and consolidate diverse patterns, indicating that its contextual pattern bank has adaptive inductive capabilities for various types of spatio-temporal patterns, independent of the specific dataset type. This mechanism enables STBP to exhibit greater flexibility and adaptability in CSTF tasks.

In addition, Figure 13 provides an intuitive comparison of the predictive performance of STBP and the second-best model, EAC, in real-world application scenarios. These representative cases further substantiate the superior practical utility of STBP in realistic continual learning settings.

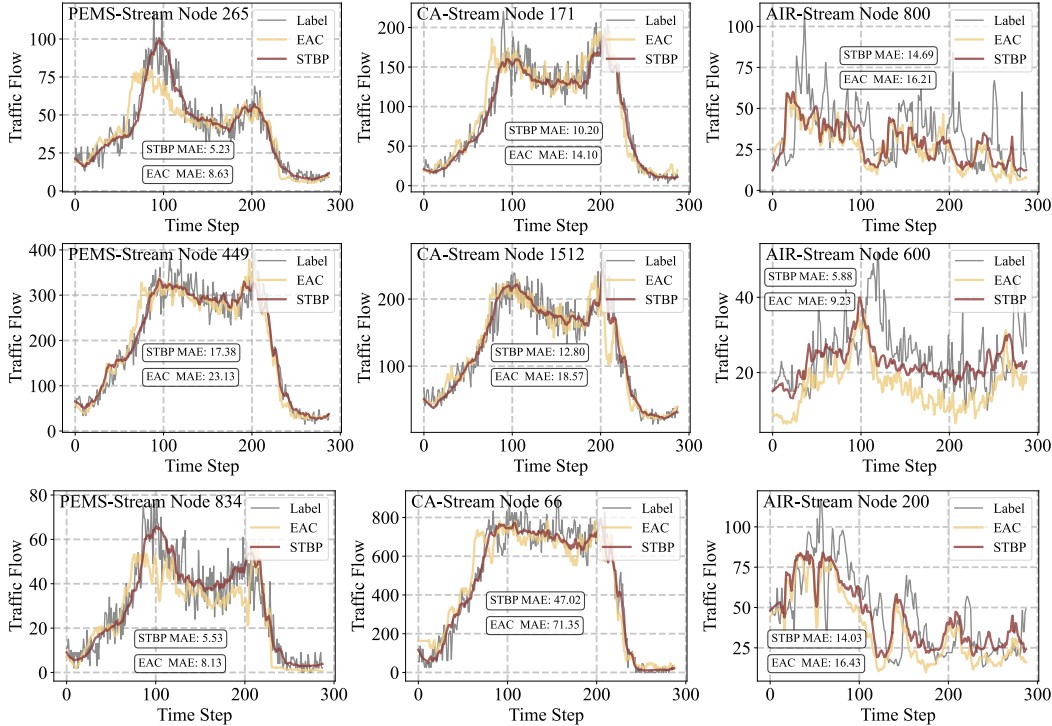

Figure 13: Additional visualization of real forecasting results.

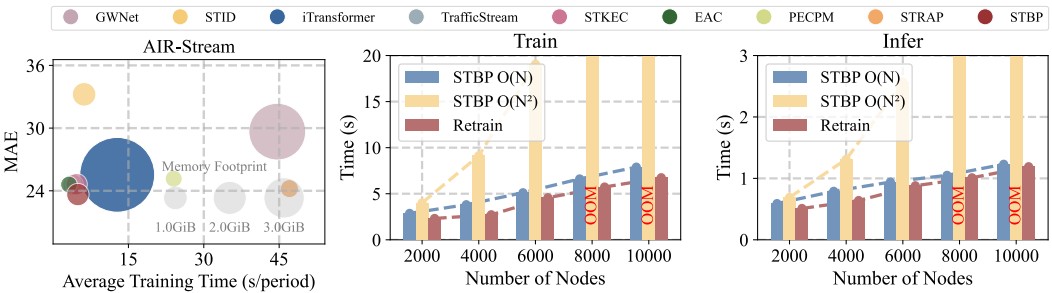

Figure 14: Additional efficiency comparison. STBP $O(N^2)$ denotes the version without linear attention, and Retrain refers to removing the contextual pattern bank.

### A.4.7 EFFICIENCY STUDY

Figure 14 provides additional experiments assessing the efficiency and scalability of STBP. Overall, these results confirm that STBP achieves favorable scalability and efficiency, and that its linear-attention design and modular contextual pattern bank structure enable it to handle large-scale spatio-temporal graphs in continual learning settings.

### A.5 LIMITATION

Despite STBP's strong performance on benchmark datasets, its limitations in cross-domain generalization warrant further investigation. Current continual learning approaches, including ours, typically assume incremental tasks originate from similar domains—an idealization that diverges from real-world dynamic and heterogeneous environments. In practice, cross-domain distribution shift introduce dual challenges: feature space misalignment and exacerbated catastrophic forgetting. While STBP's architecture exhibits inherent adaptability—with DLGA dynamically capturing topological variations and FreNet extracting domain-invariant frequency patterns—its robustness remains unverified under significant structural divergence between source and target domains. Future work should therefore validate the framework's efficacy in such complex cross-domain scenarios.

## A.6 BROADER IMPACT

`STBP`, with its carefully designed general spatio-temporal backbone and contextual pattern bank expansion mechanism tailored for dynamic scenario changes, effectively achieves continual spatio-temporal forecasting. This approach demonstrates that the spatio-temporal backbone can serve as a stable infrastructure, consistently retaining the ability to model general spatio-temporal dependencies. When facing new or evolving scenarios, there is no need to retrain the backbone. Instead, by introducing scalable parameters relevant to the current scenario, the model can rapidly adapt to new tasks. Building on this concept, we aim to advance the development of a spatio-temporal foundational model with enhanced cross-domain generalization, while concurrently exploring the potential of Large Language Models (LLMs) in spatio-temporal and time-series forecasting tasks (Liu et al., 2025b; 2024a; 2025a).

This involves two key directions: ❶ introducing explicit domain adaptation mechanisms to better distinguish between domain-specific and shared features, and ❷ exploring cross-domain shared contextual pattern banks to enhance adaptability while maintaining efficiency. This approach involves continuously training a unified backbone model with spatio-temporal data from multiple heterogeneous domains, thereby enhancing its spatio-temporal representational capacity. As data from various domains are continuously integrated and trained, the spatio-temporal foundational model will evolve, enabling efficient generalization and adaptation to entirely new scenarios or tasks by incorporating only a small number of additional parameters. Such a model holds the potential to benefit society by improving intelligent transportation through more accurate traffic forecasting and supporting climate resilience via advanced environmental modeling.

## A.7 LLM USAGE

In accordance with the ICLR 2026 policy on large language model (LLM) usage, we disclose that we used an LLM (ChatGPT) solely for the purpose of improving the grammar, clarity, and fluency of the manuscript. The content, structure, technical contributions, experiments, analysis, and all scientific writing were entirely conceived, drafted, and validated by the human authors. The LLM was not involved in research ideation, experimental design, data analysis, or any aspect of the scientific content creation. All outputs generated by the LLM were reviewed and edited by the authors to ensure accuracy and correctness. We confirm that no hidden prompts, prompt injections, or LLM-generated falsehoods were introduced in the manuscript, and all use of LLMs complies with the ICLR Code of Ethics.

