# OpenReview forum: "A General Spatio-Temporal Backbone with Scalable Contextual Pattern Bank for Urban Continual Forecasting"
_ICLR.cc/2026/Conference — ICLR 2026 Poster_

### Official Review · Reviewer_qSGs · 2025-10-25

**Soundness:** 3
**Presentation:** 4
**Contribution:** 2
**Rating:** 6
**Confidence:** 4

**Summary:**

The paper proposes STBP, a framework that aims to unify STGNNs with continual learning. By combining a general spatio-temporal backbone and a scalable contextual pattern bank, the method addresses challenges of node expansion and concept drift through prompt-guided adaptation. Experiments on three public benchmark datasets show that STBP achieves competitive accuracy while maintaining efficiency and scalability.

**Strengths:**

1. Provides a general formulation for streaming spatio-temporal graphs that can be extended beyond traffic applications, with clear practical relevance to smart cities.
2.  Conducts comprehensive experiments with diverse presentation formats, supporting reproducibility and interpretability.
3.   Well-organized and clearly written.

**Weaknesses:**

1. The proposed method integrates multiple components, e.g., frequency-domain network, dual-stream linear graph attention, prompt-based continual learning, but each component is relatively straightforward and has been explored in related domains. The overall algorithmic novelty is limited.
2.  While the authors propose a model-agnostic continual learning strategy, they do not demonstrate its generality across different backbone architectures, leaving its broader effectiveness unverified.

**Questions:**

1.  Figure 3 shows well-separated clusters in the contextual pattern bank $ P_{\tau} $, yet the method does not include explicit constraints to enforce such separation. Could the authors clarify what drives this differentiation?
2.  Compared to similar prompt-tuning approaches such as EAC, what are the specific advantages and contributions of the contextual pattern bank?
3. The output of FreNet is $ H_{\tau}^f $. How is it transformed so that the input of DLGA is $ H_{\tau}^s $? The author did not mention it in the paper.

---

> ### Author Response · Authors · 2025-11-21
> **Response to Reviewer qSGs**
>
> Dear Reviewer qSGs,
>
> We sincerely appreciate your thorough review and valuable comments. Below, we provide a point-by-point response to each of your questions and concerns.
>
> ---
>
> **Response to W1:**
>
> Thank you for this important feedback. While individual components in STBP have research foundations in related fields, our core contribution lies in the **systematic integration** specifically designed to address key challenges in Continual Spatio-Temporal Forecasting (CSTF).
>
> Most existing CSTF methods adopt simplistic backbones (e.g., "GCN + 1D-CNN") and underemphasize spatio-temporal modeling in incremental scenarios. STBP redesigns the backbone to directly address this:
>
> + **FreNet** extracts stable components (e.g., periodicity, trends) via frequency-domain modeling to alleviate temporal distribution drift.
> + **DLGA** models dynamic spatial correlations in a data-driven manner, adapting to node expansion without predefined graphs.
>
> Ablation studies confirm the backbone's effectiveness: using it alone for incremental training achieves strong performance **(PEMS-Stream MAE: 12.52)**, outperforming EAC **(MAE: 13.49)**.
>
> Beyond backbone redesign, we emphasize **deep collaboration** between the pattern bank and backbone. The pattern bank's three component groups interact with hidden layers and attention modules via gating and key-based mechanisms, enhancing expressiveness without significant overhead. Interpretability analysis and experiments validate this design.
>
> In summary, STBP's novelty lies in its **holistic framework** that systematically balances stability, adaptability, and interpretability for CSTF. We have clarified this integrated design philosophy in the revised manuscript.
>
> ---
>
> **Response to W2:**
>
> Thank you for raising this point. We clarify that the pattern bank is designed for **deep collaboration** with the backbone—not full independence. Specifically, $\mathbf{P}\_\tau^{(2)}$ serves as attention keys in the DLGA module, directly guiding correlation modeling.
>
> The term "agnostic" refers to the backbone's **scenario independence**: it requires no predefined graph structures and has node-count-independent parameters, enabling flexible adaptation to various graph settings.
>
> To demonstrate adaptability, we conducted an experiment replacing STBP's backbone with EAC's STGNN structure while retaining a simplified pattern bank (only gating components $\mathbf{P}\_\tau^{(0)}$ and $\mathbf{P}\_\tau^{(1)}$). Results show this hybrid still outperforms original EAC:
>
> **Performance of STBP with Generic Backbone vs. EAC:**
> >
> > | Dataset | Metric | STBP w STGNN* | EAC |
> > |:---:|:---:|:---:|:---:|
> > | PEMS-Stream | MAE | 12.93±0.09 | 13.49±0.15 |
> > | | RMSE | 21.09±0.15 | 21.60±0.19 |
> > | | MAPE | 17.13±0.29 | 19.69±0.90 |
> > | CA-Stream | MAE | 17.15±0.03 | 17.91±0.24 |
> > | | RMSE | 27.01±0.09 | 27.73±0.25 |
> > | | MAPE | 16.23±0.19 | 17.49±0.24 |
> > | AIR-Stream | MAE | 20.66±0.30 | 20.77±0.24 |
> > | | RMSE | 32.78±0.43 | 32.94±0.34 |
> > | | MAPE | 25.84±0.26 | 26.77±0.39 |
>
> _*STBP w STGNN uses EAC's backbone architecture._
>
> This confirms the pattern bank's portability and cross-backbone effectiveness. We have added relevant explanations in the revised manuscript to clarify this point.

---

> ### Author Response · Authors · 2025-11-21
> **Response to Reviewer qSGs**
>
> **Response to Q1:**
>
> Thank you for this question. The clustering emerges **naturally** without explicit constraints, driven by:
>
> + **Data-Driven Representation Learning:** The pattern bank $\mathbf{P}\_\tau \in \mathbb{R}^{N\_\tau \times d}$ consists of trainable parameters optimized via prediction loss (Eq. 2). This encourages nodes with similar dynamics to converge in parameter space, while dissimilar nodes diverge, autonomously forming clusters.
> + **Structured Prompt Interactions:** The three parameter groups ($\mathbf{P}\_\tau^{(0)}$, $\mathbf{P}\_\tau^{(1)}$, $\mathbf{P}\_\tau^{(2)}$) interact with the backbone via gating and attention mechanisms (Eq. 5), guiding the model to adaptively distinguish node similarities and differences.
>
> Thus, the observed clusters are semantically meaningful representations formed through task-driven learning and structured interactions. We will emphasize this point in Section 4.2 of the revised manuscript.
>
> ---
>
> **Response to Q2:**
>
> Thank you for your question. Unlike the "expand-and-compress" dynamic prompt pool used in EAC, our contextual pattern bank offers two key advantages in continual learning mechanism and structural design:
>
> + **Incremental Expansion Without Compression, Avoiding Historical Information Loss:** EAC's prompt pool requires a compression mechanism after expansion to control its size, which may lead to the loss of previously learned patterns. In contrast, our pattern bank adopts a **purely parametric incremental expansion** strategy (only adding $\Delta \mathbf{P}\_\tau$), avoids compressing existing parameters, and integrates with the backbone via a linear interaction strategy. Even under drastic node increases (e.g., +254% in CA-Stream), this does not cause exponential growth in computational cost while more completely preserving historical knowledge. Experimental results show that this approach reduces MAE by **16.69%** compared to EAC.
> + **Joint Modeling of Node Relevance and Heterogeneity, Enhancing Expressiveness:** EAC's prompt pool is primarily used to adapt to new nodes, and its interaction with the backbone is simple (e.g., feature addition), failing to explicitly distinguish between common patterns and individual node differences. Our pattern bank divides parameters into three groups—$\mathbf{P}\_\tau^{(0)}$, $\mathbf{P}\_\tau^{(1)}$, and $\mathbf{P}\_\tau^{(2)}$—which participate in **gating and attention mechanisms**, respectively. This enables the model to simultaneously capture node-specific patterns (**heterogeneity**) and shared patterns among nodes (**relevance**), leading to more comprehensive spatio-temporal representation learning. We also provide interpretability analysis to explain how the pattern bank functions.
>
> In summary, the contextual pattern bank, through its **compression-free expansion mechanism** and **structured multi-component design**, achieves superior information retention and pattern modeling capabilities during continual learning. We have emphasized the above content in the revised manuscript.
>
> ---
>
> **Response to Q3:**
>
> Thank you for pointing this out. The transformation is as follows:
>
> FreNet processes input $\mathbf{X}\_\tau \in \mathbb{R}^{N\_\tau \times T\_h}$ via:
> `Linear Layer → FFT → Frequency Embedding → IFFT → Linear Layer`
> to output $\mathbf{H}^{f}\_{\tau} \in \mathbb{R}^{N\_{\tau} \times d}$ (Eq. 6). Then, $\mathbf{H}^{f}\_{\tau}$ interacts with pattern bank component $\mathbf{P}\_\tau^{(0)}$ via gating-based prompt guidance (Eq. 5) to produce the DLGA input $\mathbf{H}^{s}\_{\tau} \in \mathbb{R}^{N_{\tau} \times d}$.
>
> Both tensors share the same dimensionality. We have added this clarification to Section 4.3 in the revised manuscript.
>
> ---
> We thank you again for your valuable feedback, which has helped us improve the clarity and rigor of our work. All revisions have been incorporated into the updated manuscript.

---

> ### Comment · Reviewer_qSGs · 2025-11-21
>
> Thanks for the response. Some problems have been solved. However, the contribution of this paper looks trival and what is the unique/large technical/theory contributions in this paper?

---

> > ### Author Response · Authors · 2025-11-22
> >
> > Thank you for your prompt reply! We understand your concerns regarding the paper's contributions and wish to further clarify the core contributions of this work at the technical, theoretical, and experimental levels.
> >
> > ---
> >
> > **Technical Contribution 1: A Novel General Spatio-Temporal Backbone Designed for Continual Learning**
> >
> > We propose a novel framework named **STBP**, whose core is a spatio-temporal backbone specifically designed to address the core challenges of continual learning. It comprises two key innovative components:
> >
> > - **Frequency-Domain Network (FreNet)**: To tackle the issue of data distribution drift in continual learning, we move beyond traditional temporal modules like RNNs/TCNs/Attention and design a frequency-domain processing module. This module more effectively isolates and reinforces stable periodic components and trends in the data, thereby enhancing the model's inherent robustness to gradual distribution shifts. Concurrently, this design also offers significant computational efficiency advantages over traditional temporal models.
> > - **Dual-stream Linear Graph Attention (DLGA)**: To address the computational complexity challenge arising from the continuous expansion of graph structures in continual learning, we propose the **Dual-stream Linear Graph Attention**. It reduces the quadratic computational complexity of traditional attention mechanisms to *linear* complexity through random feature mapping, enabling it to handle large-scale dynamic graphs. More importantly, this mechanism does not require pre-defined graph structures, can capture dynamic spatial correlations in a fully data-driven manner, and innovatively introduces a "prompt stream" driven by the contextual pattern bank, allowing the model to consider both the intrinsic relationships within the input data and their relevance to historical knowledge during inference.
> >
> > ---
> >
> > **Technical Contribution 2: A Continual Learning Optimization Strategy Targeting the Stability-Plasticity Trade-off**
> >
> > To address the challenges of catastrophic forgetting and graph structure expansion within the continual learning framework, we design a **scalable contextual pattern bank**.
> >
> > - This pattern bank dynamically stores node-relevant heterogeneous contextual patterns through **parametric incremental expansion**.
> > - Via a **linear-complexity prompt guidance mechanism**, it efficiently collaborates with the frozen backbone network, enabling rapid adaptation to new scenarios without compromising historical knowledge.
> > - This design not only elegantly balances stability and plasticity in its mechanism but also provides a degree of interpretability for model decisions through its pattern clustering characteristics.
> >
> > ---
> >
> > **Theoretical Contribution: A Unified Decoupled Learning Framework and Efficiency Guarantees**
> >
> > At the theoretical level, the contributions of this work are:
> >
> > - **A Decoupled Learning Framework**: We explicitly separate the learning of *stable, general spatio-temporal patterns* (handled by the backbone) from the adaptation to *context-specific patterns* (handled by the contextual pattern bank), providing a clear theoretical foundation for continual spatio-temporal learning.
> > - **Rigorous Efficiency and Scalability Guarantees**: Through the linear attention mechanism and the lightweight interaction design of the pattern bank, we ensure that the overall computational and memory complexity of STBP grows *linearly* with the number of nodes. We provide not only rigorous theoretical complexity analysis but also validate its feasibility for deployment in large-scale scenarios through systematic efficiency experiments.
> >
> > ---
> >
> > **Experimental Contribution: Exceptional Robustness and Generalization in Extreme Scenarios**
> >
> > At the exceptional level, STBP demonstrates strong robustness in scenarios beyond conventional settings, including:
> >
> > - **Few-shot learning** (e.g., with **90%** of the training samples missing)
> > - **Extreme incremental tests** (e.g., explosive node growth of **+254%**)
> >
> > Under *all* challenging settings, STBP significantly and consistently outperforms all baseline models. For instance, compared to the best baseline, STBP reduces the average MAE by **18.46%**, **16.37%**, and **5.1%** on the three mainstream datasets, respectively.
> >
> > ---
> >
> > We believe these contributions are both unique and significant. They not only represent a further exploration beyond existing methods but also provide a **high-performance, scalable, and interpretable solution** for spatio-temporal forecasting in real-world dynamically evolving environments. Thank you again for your feedback.

---

> > > ### Comment · Reviewer_qSGs · 2025-11-25
> > >
> > > Thanks for the response. I do not have further comments.

---

> > > > ### Author Response · Authors · 2025-11-26
> > > >
> > > > Thank you again for your constructive feedback, which have been very helpful in improving the paper. We also appreciate your efforts and wish you all the best.

---

### Official Review · Reviewer_3Woj · 2025-10-27

**Soundness:** 4
**Presentation:** 4
**Contribution:** 3
**Rating:** 8
**Confidence:** 5

**Summary:**

This paper addresses the challenge of modeling dynamic and evolving spatio-temporal data in urban environments and proposes STBP, a continual forecasting framework. The method combines a general spatio-temporal backbone with a scalable contextual pattern bank, aiming to handle distributional drift and topological evolution without retraining the entire model. Extensive experiments conducted on three real-world streaming datasets demonstrate that STBP outperforms existing methods in terms of predictive accuracy, scalability, and resistance to catastrophic forgetting.

**Strengths:**

- Originality: The paper introduces a novel combination of frequency-domain modeling, lightweight linear graph attention, and a scalable prompt-based contextual parameter expansion strategy. This bridges the gap between spatio-temporal modeling and continual learning, making a meaningful contribution to the continual spatio-temporal forecasting field.

- Quality: The model architecture is well-structured with clear theoretical underpinnings. Each module is logically integrated to form a coherent pipeline. Specific designs address key challenges in CSTF, such as distributional drift, topological changes, and catastrophic forgetting.

- Clarity: The experimental setup is comprehensive, covering multiple representative real-world datasets. The appendix provides additional implementation details, which enhance reproducibility and interpretability.

- Significance: The proposed framework has practical value and potential applicability in domains such as traffic forecasting and meteorological analysis, where long-term adaptation to evolving topology and data distributions is essential.

**Weaknesses:**

1. Although the paper briefly acknowledges the challenge of cross-domain generalization, the main text lacks a detailed analysis of how STBP handles domain shifts, especially when there are significant structural differences between source and target tasks.

2. The current design tightly couples the contextual pattern bank with the backbone, which may limit the modularity of the backbone and restrict its transferability to other domains or tasks.

3. There is a typo in the title of Figure 5: "PESM-Stream" should be corrected to "PEMS-Stream".

**Questions:**

1. During parameter expansion of the contextual pattern bank, have redundancy or performance degradation issues been observed?

2. The use of frequency-domain modeling is claimed to help mitigate distributional drift. From a theoretical perspective, what are the advantages of this approach?

---

> ### Author Response · Authors · 2025-11-21
> **Response to Reviewer 3Woj**
>
> Dear Reviewer 3Woj,
>
> We sincerely appreciate your insightful comments and constructive suggestions. Below, we provide a point-by-point response to each of your concerns.
>
> ---
>
> **Response to W1:**
>
> Thank you for this insightful comment. We fully agree that cross-domain generalization is an important and valuable direction for deeper exploration.
>
> While our current experiments focus primarily on single-domain continual learning, the architectural design of STBP inherently possesses potential for cross-domain adaptation:
>
> + The **DLGA module** dynamically models node correlations in a data-driven manner, enabling flexible adaptation to varying topological structures across domains.
> + The **frequency-domain features** (e.g., low-frequency components like periodicity and trends) extracted by the FreNet module often exhibit cross-domain commonalities, helping mitigate distribution shifts.
>
> We acknowledge that systematic cross-domain analysis remains a limitation of the current work. In future research, we plan to advance in two key directions:
>
> + Introducing explicit domain adaptation mechanisms to better distinguish between domain-specific and shared features.
> + Exploring cross-domain shared contextual pattern banks to enhance adaptability while maintaining efficiency.
>
> We have added discussions on this cross-domain potential and future directions in Section A.6 of the revised manuscript.
>
> ---
>
> **Response to W2:**
>
> Thank you for raising this important point. The collaborative design between the pattern bank and backbone represents a deliberate trade-off tailored to continual spatio-temporal forecasting. Our goal is to balance stable knowledge retention and dynamic scenario adaptation:
>
> + The **frozen backbone** preserves general spatio-temporal modeling capabilities.
> + The **contextual pattern bank** dynamically injects scenario-specific information via prompt mechanisms, enabling efficient adaptation while avoiding potential information loss.
>
> Importantly, this design maintains modularity and transferability:
>
> + The backbone, as an independent structure, possesses general modeling capabilities and can be directly transferred across tasks (e.g., traffic → air quality forecasting).
> + The pattern bank stores scenario-related parameters in a lightweight manner, supporting incremental expansion and independent optimization.
>
> Thus, STBP forms a modular architecture of **"general backbone + specialized pattern bank"**, maintaining cross-task generality while ensuring adaptation efficiency. We have clarified these points in the revised manuscript.
>
> ---
>
> **Response to W3:**
>
> Thank you for your careful review. We have corrected the typo in the caption of Figure 5 from "PESM-Stream" to "PEMS-Stream" in the revised manuscript.

---

> ### Author Response · Authors · 2025-11-21
> **Response to Reviewer 3Woj**
>
> **Response to Q1:**
>
> Thanks for raising this question. In our design and experiments, we did not observe significant redundancy or performance degradation from pattern bank expansion. Key reasons include:
>
> + **Data-Driven Expansion with Linear Interaction Strategy:** New parameters are allocated solely upon the introduction of new nodes, effectively avoiding unnecessary redundancy. Furthermore, the contextual pattern bank interacts with the backbone exclusively through linear strategies, which prevents exponential growth in computational costs. Efficiency experiments (Fig. 8) confirm that the computational overhead increases approximately linearly (~20%–30% for memory and time).
> + **Stress-Test Validation:** The stability of this expansion mechanism is rigorously validated under extreme conditions. On the CA-Stream dataset, which involves a substantial 254% node growth, STBP not only maintains stable performance across incremental stages but also significantly outperforms all compared models (Table 8), demonstrating remarkable scalability and robustness.
>
> These results indicate that the pattern bank achieves effective capacity expansion while maintaining computational efficiency. We have emphasized this point in Section 5.5 of the revised manuscript.
>
> ---
>
> **Response to Q2:**
>
> Thank you for your valuable feedback. Frequency-domain modeling offers dual advantages in handling distributional drift and improving computational efficiency:
>
> + **Representation Stability:** Core patterns (e.g., daily traffic cycles, seasonal trends) manifest as stable low-frequency components. FreNet reinforces these while suppressing high-frequency noise, yielding more robust temporal representations.
> + **Adaptability to Drift:** When time-domain statistics change, the underlying frequency structure often remains stable, enabling partial "decoupling" from statistical shifts.
> + **Computational Efficiency:** Frequency transformations offer superior time complexity versus traditional RNNs/CNNs. Combined with linear attention, this reduces long-sequence modeling costs, making it suitable for continual learning.
>
> In summary, frequency-domain modeling provides both theoretical and practical benefits for distributional robustness and efficiency in continual spatio-temporal learning. We have added this point in Section 4.3 of the revised manuscript.
>
> ---
>
> We thank you again for your valuable feedback, which has helped us improve the manuscript.

---

> > ### Comment · Reviewer_3Woj · 2025-11-26
> > **Response to rebuttal.**
> >
> > Thank you for the detailed rebuttal. One question remains: in the few-shot setting, which factors enable STBP to outperform the baselines despite limited training data?

---

> > > ### Author Response · Authors · 2025-11-26
> > >
> > > Thank you for your question. The superior performance of STBP in few-shot scenarios stems from its key designs:
> > >
> > > - **The Contextual Pattern Bank employs a lossless parameter expansion strategy**. Unlike the "expansion-compression" approach used by models such as EAC, STBP utilizes a purely incremental expansion that fully preserves historical knowledge, thereby avoiding information loss. This allows STBP to fully reuse existing historical spatio-temporal patterns in data-scarce situations, enabling rapid adaptation to new nodes.
> > >
> > > - **The General Backbone provides stable feature extraction**. Specifically, the FreNet module extracts stable components that are robust to distribution drift through frequency domain analysis, allowing the model to capture key temporal patterns with minimal data and reduce overfitting. The DLGA module dynamically models spatial dependencies with linear complexity and enhances knowledge integration via prompts from the pattern bank, ensuring efficient capture of spatial correlations during node expansion. With its stable representation capacity, this backbone inherently excels over existing methods.
> > >
> > > These designs enable STBP to retain historical knowledge while quickly adapting to new scenarios, maintaining strong performance even under few-shot conditions.

---

> > > > ### Comment · Reviewer_3Woj · 2025-11-27
> > > > **Thanks for the authors’ response.**
> > > >
> > > > Thank you for the detailed response. My concerns have been addressed. I also reviewed the authors’ replies to the other reviewers, which are clear and well-argued. Overall, I believe the authors have responded to the reviewers’ concerns effectively.

---

> > > > > ### Author Response · Authors · 2025-11-27
> > > > >
> > > > > We sincerely appreciate your positive feedback and are pleased that your concerns have been addressed. Thank you for acknowledging our work, and we wish you all the best.

---

### Official Review · Reviewer_j7eh · 2025-11-01

**Soundness:** 3
**Presentation:** 3
**Contribution:** 2
**Rating:** 4
**Confidence:** 3

**Summary:**

This paper proposes the STBP framework for urban continual spatio-temporal forecasting. The method combines a general spatio-temporal backbone with a scalable contextual pattern bank, adapting to dynamically evolving spatio-temporal data through incremental pattern bank expansion while freezing the backbone network. This design effectively balances catastrophic forgetting mitigation, dynamic correlation modeling, and computational efficiency.

**Strengths:**

1. The task of continual spatio-temporal forecasting is interesting and has significant practical value for real-world applications.

2. The proposed method demonstrates substantial performance advantages over the compared baselines across multiple datasets.

**Weaknesses:**

1. While the application scenario is interesting, the proposed method lacks significant insight. The approach of freezing the backbone network while expanding a dynamic pattern bank to handle incremental node expansion appears to be a straightforward solution, and similar strategies have been proposed in prior work (e.g., EAC uses prompt pool expansion).

2. Insufficient baseline comparisons: Only six baselines are compared. More detailed experimental settings explaining how conventional spatio-temporal forecasting models (GWNet, STID, iTransformer) are adapted for this incremental scenario would help readers better understand the main results.

3. The paper does not address a critical question: how much training data do existing baseline models require on new nodes to achieve acceptable performance? If baselines can reach satisfactory accuracy with only a short period of data (e.g., one week), the practical utility of the proposed continual learning approach would be significantly diminished. Conversely, if long-term data is necessary, this perspective could strengthen the motivation and method presentation.

**Questions:**

Please refer to Weaknesses

---

> ### Author Response · Authors · 2025-11-21
> **Response to Reviewer j7eh**
>
> Dear Reviewer j7eh,
>
> We sincerely thank the reviewer for the insightful comments and constructive suggestions. Below, we provide a point-by-point response to each concern.
>
> ---
>
> **Response to W1:**
>
> Thank you for this valuable feedback. We acknowledge that excellent works such as EAC have inspired our study. However, our key contribution lies not only in proposing an effective continual forecasting framework but, more importantly, in identifying and systematically addressing two critical issues that are widely overlooked in current research on incremental node expansion:
>
> 1. **Rethinking and Improving the Spatio-Temporal Backbone**
>    Existing CSTF methods (e.g., EAC) often adopt structurally simple "general-purpose" spatio-temporal backbones (e.g., stacks of graph and 1D convolutions), failing to incorporate insights from well-established STGNNs (e.g., GWNet, STID). This is primarily because such STGNNs rely on fixed graph topologies and are difficult to adapt to scenarios with continuously growing nodes. Modifying them for incremental learning often leads to severe performance degradation.
>
>    We argue that existing backbones suffer from two key limitations:
>     - Inability to fully model dynamic spatial correlations;
>     - Inefficiency in handling distributional drift during incremental learning.
>
>    To address these, **STBP redesigns the general-purpose backbone**:
>     - **FreNet** extracts periodic and trend components in the time dimension, alleviating temporal distributional drift;
>     - **DLGA** models dynamic spatial correlations in a data-driven manner without predefined adjacency matrices, enhancing generality and adaptability.
>
>    As shown in the table below, even when using only the backbone for online training (denoted as **Online**), STBP (**MAE:12.52/16.66**) outperforms EAC (**MAE:13.49/17.91**) on traffic tasks.
>
> 2. **Deep Collaboration Mechanism Between the Contextual Pattern Bank and the Backbone**
>    Although expandable prompt pool structures have been introduced in works like EAC, their collaboration with the backbone is relatively simple (e.g., vector addition), with limited exploration of interpretability and efficiency. Through experiments (Fig. 7), we found that the expansion of STBP's contextual pattern bank incurs only approximately linear overhead (20%–30%), without exponential cost growth. This shifted our focus from "expansion cost" to "collaboration mechanism" and "interpretability".
>
>    Specifically, STBP designs **three groups of components** in the pattern bank:
>     - The first two interact with hidden layers via gating signals to capture node heterogeneity;
>     - The third group serves as attention keys to guide node correlation modeling.
>
>    Interpretability analysis shows that these patterns clearly distinguish different node clusters in the representation space. As illustrated in the table below, thanks to tighter module collaboration, STBP significantly outperforms EAC on multiple streaming benchmarks without substantial computational cost increase.
>
> **Performance Comparison (STBP vs. Online vs. EAC):**
>
> >| Dataset | Metric | STBP | Online* | EAC |
> >|:---:|:---:|:---:|:---:|:---:|
> >| PEMS-Stream | MAE | **11.00±0.04** | 12.52±0.07 | 13.49±0.15 |
> >| | RMSE | **18.15±0.06** | 20.18±0.11 | 21.60±0.19 |
> >| | MAPE | **14.52±0.09** | 22.38±1.69 | 19.69±0.90 |
> >| CA-Stream | MAE | **14.92±0.05** | 16.66±0.43 | 17.91±0.24 |
> >| | RMSE | **24.17±0.09** | 26.35±0.49 | 27.73±0.25 |
> >| | MAPE | **14.41±0.42** | 18.66±0.93 | 17.49±0.24 |
>
> _* **Online** denotes STBP with the contextual pattern bank removed, initialized with weights from the previous period at each incremental stage._
>
> In summary, while inheriting expansion strategies from prior works, STBP introduces systematic innovations in **backbone structure design** and **module collaboration mechanisms**, addressing key issues commonly overlooked in current research. We have re-emphasized these contributions in the Introduction section of the revised manuscript.

---

> ### Author Response · Authors · 2025-11-21
> **Response to Reviewer j7eh**
>
> **Response to W2:**
>
> Thank you for this suggestion. We have enhanced our revision in the following aspects:
>
> + **Additional Baselines**: We have included two more CSTF methods—**PECPM** [1] (KDD'23) and **STRAP** [2] (NeurIPS'25) —to enrich the comparison. We also supplemented corresponding computational efficiency analyses, confirming that STBP remains highly competitive.
> + **Adaptation of Conventional STGNNs**: Models like **GWNet** and **STID** have parameter sizes strongly tied to the number of nodes and rely on fixed graph topologies. To simulate their performance in incremental settings, we **retrain the backbone from scratch** at each stage using only the current period's data, without leveraging historical knowledge.
> + **Adaptation of iTransformer**: Since iTransformer's parameters are node-count-agnostic, we adopt an **online training** regime: at each stage, the model is initialized with weights from the previous stage and fine-tuned on the full node data of the current stage, thereby utilizing learned spatio-temporal knowledge.
>
> We have updated the relevant experimental results and settings in the Experiments section of the revised manuscript.
>
> **References:**
>
> [1] Wang, et al. "Pattern Expansion and Consolidation on Evolving Graphs for Continual Traffic Prediction." SIGKDD, 2023.
>
> [2] Zhang, et al. "STRAP: Spatio-Temporal Pattern Retrieval for Out-of-Distribution Generalization." NeurIPS, 2025.
>
> ---
>
> **Response to W3:**
>
> Thank you for raising this critical point. We fully agree: if baseline models require only a small amount of data to achieve good performance, the necessity of continual learning would be greatly reduced. In practice, data requirements vary significantly across scenarios, primarily depending on their periodicity and pattern complexity:
>
> + **Data Requirements in Real-World Scenarios**:
>     - **Traffic Flow Forecasting (e.g., PEMS)**: Data sampled at 5-minute intervals exhibits clear daily and weekly periodicities. Typically, **about one month** of training data is needed for stable performance;
>     - **Air Quality Forecasting (e.g., Air-Stream)**: Hourly data with seasonal variations often requires **at least one year** of data to cover a full cycle and achieve acceptable performance.
> + **Few-Shot Experiment**:
>   To validate baseline adaptability under limited data, We have added a few-shot learning experiment in Section 5.2 of the revised manuscript. The setup is as follows:
>     - Only the first incremental stage uses the full dataset; Each subsequent stage retains only **10%** of the original training samples, while the test set remains unchanged;
>     - In PEMS-Stream and CA-Stream, this equates to **~2–3 days of training data per new node**.
> + **Results and Analysis**:
>   The results below show that even under extreme data scarcity, CSTF methods (e.g., STBP, EAC) significantly outperform conventional baselines. This is because:
>     - Conventional models struggle to capture stable spatio-temporal patterns with limited data;
>     - CSTF methods leverage knowledge from historical stages to adapt quickly to new nodes;
>     - The continual learning mechanism mitigates catastrophic forgetting, enabling continuous use of previously learned features.
>
> Thus, our method demonstrates clear practical value in real-world scenarios where data is often insufficient.
>
> **Few-Shot Forecasting Performance (10% Training Data):**
>
> | Model | PEMS-Stream 10% MAE | CA-Stream 10% MAE | AIR-Stream 10% MAE |
> |-------|---------------------|------------------|-------------------|
> | GWNet | 25.64±0.38 | 34.44±0.46 | 36.66±1.15 |
> | STID | 29.14±0.09 | 36.90±0.57 | 43.89±0.49 |
> | iTransformer | 18.53±0.54 | 22.14±0.23 | 27.60±0.20 |
> | TrafficStream | 15.04±0.08 | 18.92±0.38 | 25.10±0.63 |
> | STKEC | 15.54±0.13 | 18.60±0.28 | 24.14±0.88 |
> | PECPM | 14.81±0.07 | 18.78±0.26 | 24.10±0.38 |
> | STRAP | 15.44±0.13 | 24.49±0.85 | 24.94±1.32 |
> | EAC | 14.85±0.12 | 18.52±0.44 | 25.48±0.48 |
> | STBP | **12.85±0.06** | **16.17±0.05** | **22.57±0.76** |
>
> ---
>
> We hope these revisions and clarifications adequately address your concerns. Thank you again for your valuable input.

---

> ### Author Response · Authors · 2025-11-27
>
> Dear Reviewer j7eh,
>
> We are hopeful that the revisions, including the new experiments and clarifications, have adequately addressed the points you raised. If these changes meet with your approval, we would be most appreciative if you would consider updating your score accordingly. We remain fully available to discuss any remaining issues.
>
> Best regards,
>
> All Authors

---

> > ### Comment · Reviewer_j7eh · 2025-11-28
> >
> > I appreciate the author's detailed response. The additional experiments have successfully resolved the majority of my concerns, particularly regarding points 2 and 3. As a result, I am inclined to increase my final score to the range of 5 to 6.

---

> > > ### Author Response · Authors · 2025-11-28
> > >
> > > We sincerely thank you for raising your score and for your invaluable contributions, which have greatly improved our manuscript. We wish you all the best.

---

### Author Response · Authors · 2025-11-21
**General Response**

We extend our sincere gratitude to all reviewers for their invaluable time and insightful comments, which have significantly contributed to improving our manuscript. We appreciate the opportunity to address the points raised by **Reviewers j7eh, 3Woj, and qSGs**, and have revised the paper accordingly. Below we summarize the key improvements made in response to their feedback:

+ **To Reviewer j7eh:** We have reorganized the introduction to better highlight our contributions, introduced two additional baselines (PECPM and STRAP) with full analysis in the main text and appendix, and moved the few-shot experiments from the appendix to the main body to strengthen the evaluation.
+ **To Reviewer 3Woj:** We have added a dedicated discussion in the appendix on limitations and future work—including cross-domain generalization—and clarified the transferability of the backbone design. Typographical errors have been corrected, efficiency analyses regarding redundancy have been reinforced, and the methodological advantages of FreNet have been elaborated in the corresponding section.
+ **To Reviewer qSGs:** We have refined the introduction to better convey the integrated novelty of STBP, expanded ablation studies with discussions on scenario-agnostic design, provided further clarification in Methodology regarding pattern bank optimization and data flow from FreNet to the DGLA, and enhanced the related work to more clearly differentiate our pattern bank mechanism from existing methods.
+ **As a general improvement,** we have also enriched the main text and appendix with new case studies that include visual forecasting analysis to further support our empirical claims.

All revisions have been incorporated into the updated manuscript, with all modifications clearly highlighted to facilitate your review. We are confident that this revision process has significantly strengthened our work. Should you have any further questions, we are fully prepared to provide detailed clarifications. Once again, we extend our sincere gratitude for your invaluable insights and constructive guidance.

---

### Meta-Review · Area_Chair_7Mju · 2026-01-05

**Summary:**

This paper proposes a spatio-temporal backbone for continual forecasting framework. This addresses the challenges of node expansion and concept drift through prompt-guided adaptation.
The backbone includes two components: FreNet extracts the periodic and trend components in the time dimension and DLGA captures the dynamic spatial correlations. This is complemented by a contextual pattern bank that allows scalable expansion, enabling adaptations to node expansion and distribution shift.
The reviewers have asked for additional clarification about the justification of the backbone model design, the contributions of different components, further baselines, and cross-domain adaptations. One of the reviewers also asked for clarification on novelty, and another reviewer also commented on the lack of insights provided in the paper, given that the approaches of the paper have been adopted in literature separately.
The authors have included further experiments as requested by reviewers, and have seemed to address most of the concerns. However, novelty seems to still be an issue. This can be clarified further with framing and improvement on the contribution statements.

This AC is also not fully convinced that the node expansion in the traffic or network graph is dealt with, as the node expansion is implicitly induced from the changing data distribution. The embedding visualization provides hints on clusters but not on the node expansion.
Node expansion is handled implicitly, not explicitly. STBP allocates new parameters in a pattern bank and uses a node-agnostic frozen backbone so practically new nodes can be processed without retraining. However, the paper treats node arrival largely as a change in data distribution rather than as a graph structural event with explicit modeling of the structural expansion of the nodes.

Therefore further revisions are required for the camera ready.
Clarification is needed. Node expansion is incorrectly used term here. Rather, paraphrase to distinguish practical ingestion/parameter expansion from explicit structural modeling of graph growth.
Further, forgetting rate after node expansion is not evaluated.

**Reviewer Concerns:**

The rebuttal has clarified the questions on additional baselines, the amount of data needed (few shot setting), and under data scarcity.
 However, the authors have not seemed to fully clarify the contribution of this paper. One of the reviewers remarked that it looks trivial.
The response to the rebuttal seem to have not fully addressed the concerns. Therefore, although the scores are already on the positive end of the borderline range, I strongly recommend the authors to revise the paper further for camera ready (if accepted), or revise it further to close the gap on the continual learning claims (if rejected)

**Reviewer Scores:**

I believe reviewer j7eh has changed their score as also stated in the rebuttal. The score has increased to 6.
I dont believe the other two have changed their scores. The scores are on the top end of the borderline range.

---

### Decision · Program_Chairs · 2026-01-26

Accept (Poster)